# Alstrom syndrome gene is a stem-cell-specific regulator of centriole duplication in the *Drosophila* testis

**Cuie Chen[1], Yukiko M Yamashita[1,2,3,4]***

[1]Life Science Institute, Department of Cell and Developmental Biology, Michigan Medicine, University of Michigan, Ann Arbor, United States; [2]Howard Hughes Medical Institute, Cambridge, United States; [3]Whitehead Institute for Biomedical Research, Cambridge, United States; [4]Department of Biology, Massachusetts Institute of Technology, Cambridge, United States

**Abstract** Asymmetrically dividing stem cells often show asymmetric behavior of the mother versus daughter centrosomes, whereby the self-renewing stem cell selectively inherits the mother or daughter centrosome. Although the asymmetric centrosome behavior is widely conserved, its biological significance remains largely unclear. Here, we show that Alms1a, a *Drosophila* homolog of the human ciliopathy gene Alstrom syndrome, is enriched on the mother centrosome in *Drosophila* male germline stem cells (GSCs). Depletion of *alms1a* in GSCs, but not in differentiating germ cells, results in rapid loss of centrosomes due to a failure in daughter centriole duplication, suggesting that Alms1a has a stem-cell-specific function in centrosome duplication. Alms1a interacts with Sak/Plk4, a critical regulator of centriole duplication, more strongly at the GSC mother centrosome, further supporting Alms1a's unique role in GSCs. Our results begin to reveal the unique regulation of stem cell centrosomes that may contribute to asymmetric stem cell divisions.

**\*For correspondence:**
yukikomy@wi.mit.edu

## Introduction

Many stem cells divide asymmetrically to generate daughter cells with distinct fates: one self-renewing stem cell and one differentiating cell. Asymmetrically dividing stem cells in several systems show stereotypical inheritance of the mother vs. daughter centrosomes, where the mother or daughter centrosome is consistently inherited by stem cells (*Conduit and Raff, 2010*; *Habib et al., 2013*; *Januschke et al., 2011*; *Wang et al., 2009*; *Yamashita et al., 2007*). This observation has provoked the idea that centrosome asymmetry in stem cells might play a critical role during asymmetric cell divisions and that stem cell centrosomes might be uniquely regulated. However, this question remains largely unanswered due to the lack of known stem-cell-specific centrosomal proteins.

*Drosophila* male germline stem cells (GSCs) divide asymmetrically by orienting their spindle perpendicular toward the hub cells, the major niche component (*Yamashita et al., 2003*; *Figure 1A*). During male GSC divisions, the mother centrosome is always located near the hub-GSC junction, whereas the daughter centrosome migrates to the other side of the cell, leading to spindle orientation perpendicular to the hub and consistent inheritance of the mother centrosome by GSCs (*Figure 1A*; *Yamashita et al., 2007*). Previously, we reported that Klp10A, a microtubule-depolymerizing kinesin of kinesin-13 family, is enriched on GSC centrosomes, but not on centrosomes of differentiating germ cells (i.e. gonialblasts (GBs) and spermatogonia (SGs)) (*Chen et al., 2016*). Klp10A is the first protein reported to exhibit stem-cell-specific centrosome localization. RNAi-mediated depletion of *klp10A* leads to abnormal elongation of the mother centrosomes in GSCs. The abnormally elongated mother centrosome in *klp10A*-depleted GSCs result in mitotic spindles with

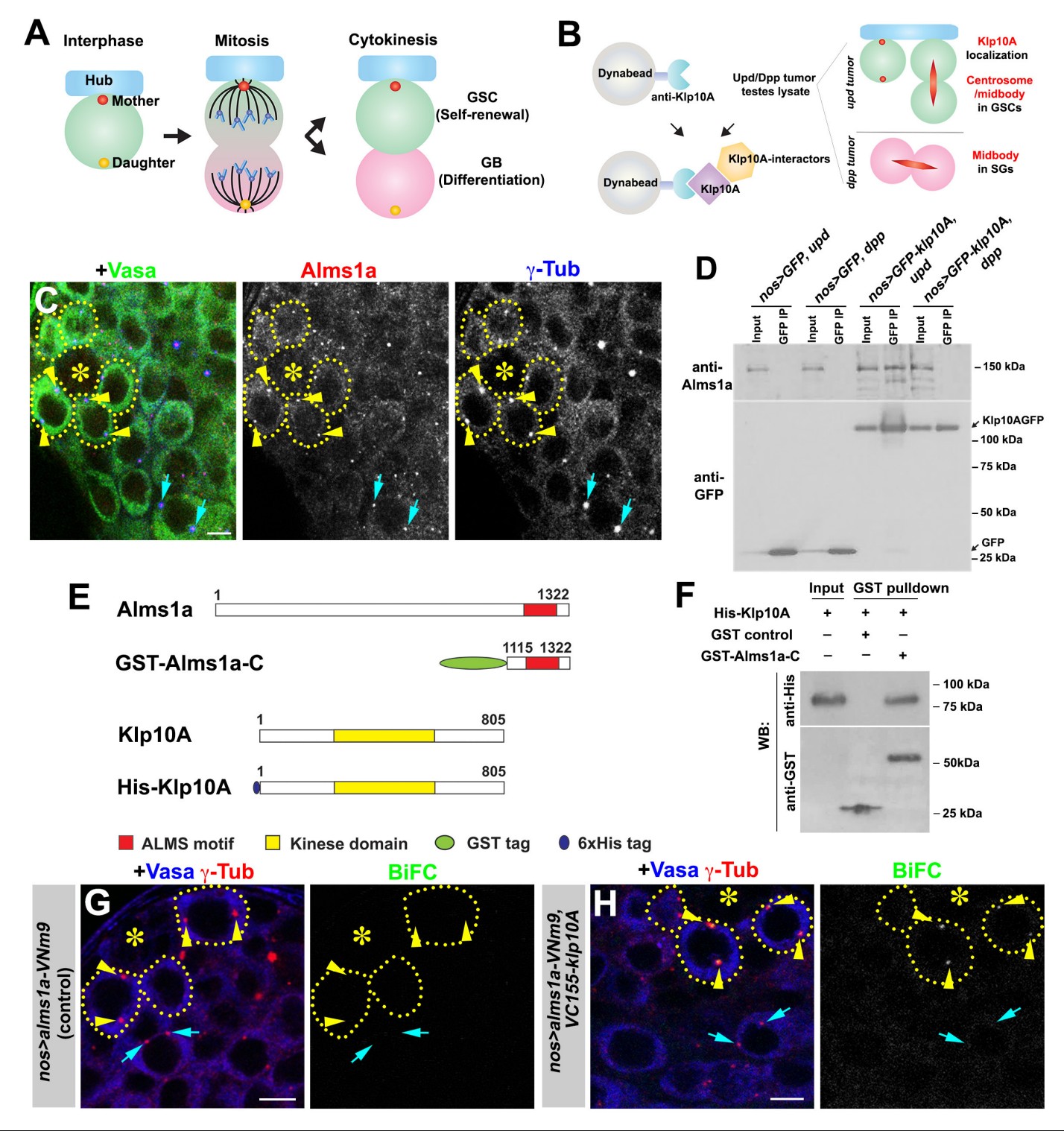

**Figure 1.** Identification of Alms1a as a germline stem cell (GSC)-specific Klp10A interactor. (**A**) Asymmetric centrosome inheritance in *Drosophila* male GSCs. (**B**) Scheme of Klp10A pulldown and mass spectrometry. The Klp10A pull down was conducted using either *upd*-induced tumor (GSC-enriched) or *dpp*-induced tumor (SG-enriched) extract, followed by mass spectrometry analysis. (**C**) An apical tip of the *Drosophila* testis stained for Alms1a (red), γ-Tub (centrosome/pericentriolar matrix, blue) and Vasa (germ cells, green). Asterisk indicates the hub. GSCs are outlined with yellow dotted lines. Arrowheads (yellow) indicate examples of GSC centrosomes. Arrows (cyan) indicate examples of SG centrosomes. Bar: 5 μm. (**D**) Co-immunoprecipitation of Klp10A and Alms1a. Control GFP and GFP-Klp10A was immunoprecipitated from GSC-enriched extracts (*nos-gal4>UAS-upd*) using an anti-GFP antibody and blotted with anti-Alms1a and anti-GFP. (**E**) Schematic of full-length Alms1a, Klp10A and constructs used for GST-

*Figure 1 continued on next page*

*Figure 1 continued*

pulldown. (F) Direct interaction of Klp10A and Alms1a C-terminal fragment. GST control and recombinant GST-Alms1a-C were immobilized on glutathione agarose beads, incubated with His-Klp10A *E. coli* lysate, and blotted with anti-GST and anti-His antibody. (G–H) Bimolecular fluorescence complementation (BiFC) analysis of Alms1a-Klp10A interaction. (G) An example of the apical tip in *nos-gal4>UAS-alms1a-VNm9* (control) testis, showing no signal. (H) An example of the apical tip in *nos-gal4>UAS-alms1a-VNm9, UAS-VC155-klp10A* testis, showing signal specifically at GSC centrosomes. Flies are raised at 18°C to minimize ectopic protein expression. Green: BiFC (Venus YFP fluorescence). Red: γ-Tub. Blue: Vasa. Arrowheads (yellow) indicate examples of GSC centrosomes positive for BiFC. Arrows (cyan) indicate examples of SG centrosomes negative for BiFC. Bar: 5 μm.

The online version of this article includes the following figure supplement(s) for figure 1:

**Figure supplement 1.** Validation of RNAi-mediated knockdown of *alms1* and antibody specificity for Alms1a and Alms1b.
**Figure supplement 2.** Bacterial two-hybrid assay showing the interaction between full-length Alms1a and full-length Klp10A.
**Figure supplement 3.** Quantification of BiFC (integrated pixel density) on centrosomes in the indicated genotypes.

an asymmetric morphology (large vs. small half spindles) and the generation of a large GSC and a small GB that often dies, likely due to an insufficient cell size (*Chen et al., 2016*). These results suggested that stem cell centrosomes are indeed under unique regulation.

To obtain further insights into the mechanism and regulation of GSC centrosome asymmetry, we aimed to identify additional proteins that exhibit GSC-specific centrosome localization by taking advantage of Klp10A's enrichment on GSC centrosomes. Here, we identify Alms1a, a *Drosophila* homolog of the causative gene for the human ciliopathy Alstrom syndrome, as a GSC-specific Klp10A-interacting protein. Human ALMS1 is reported to localize to the centrosome/basal body in many cells and function in formation/maintenance of the cilia (*Andersen et al., 2003*; *Hearn et al., 2005*). However, the underlying molecular mechanisms of its function remain poorly understood. We found that *Drosophila* Alms1a exhibits a unique asymmetric localization between mother and daughter centrosomes in GSCs, in which the GSC mother centrosomes have a higher amount of Alms1a than the GSC daughter centrosome. We found that *alms1a* depletion leads to a failure in centrosome replication, due to a failure to duplicate centrioles (the core of the centrosome), specifically in asymmetrically dividing GSCs. In contrast, *alms1a* is not required for centriole duplication in symmetrically dividing differentiating cells and GSCs induced to divide symmetrically. We show that Alms1a physically interacts with Sak, a key kinase that promotes centriole duplication, preferentially in GSCs. Taken together, our study begins to reveal the unique molecular composition and regulation of stem cell centrosome, paving the way to understand how stem cell centrosome asymmetry may contribute to asymmetric stem cell divisions.

## Results

### Identification of Alms1a as a GSC-specific Klp10A interactor

We previously reported that Klp10A, a microtubule-depolymerizing kinesin of kinesin-13 family, is enriched on GSC centrosomes, but not the centrosomes of differentiating germ cells (i.e. GBs and SGs) (*Chen et al., 2016*). We reasoned that identifying Klp10A-interacting proteins may uncover proteins that regulate the asymmetric behavior of the mother and/or daughter centrosomes in GSCs. As Klp10A localizes to the central spindle of all germ cells in addition to its GSC-specific centrosome localization (*Chen et al., 2016*; *Figure 1B*), we compared the results of anti-Klp10A immunopurification-mass spectrometry analysis using GSC-enriched extract vs. SG-enriched extract. We immunopurified Klp10A from testis extract that is enriched for either (1) GSCs due to the overexpression of the self-renewal factor, Upd (*Kiger et al., 2001*; *Tulina and Matunis, 2001*) or (2) SGs due to the overexpression of Dpp, a bone morphogenic protein (BMP) (*Bunt and Hime, 2004*; *Kawase et al., 2004*; *Schulz et al., 2004*). We reasoned that GSC centrosome-specific proteins would be enriched in the Klp10A-pulldown from GSC-enriched extracts compared to the Klp10A-pulldown from SG-enriched extracts.

Using this approach, we identified 260 candidates (*Supplementary file 1*), including two homologs of *Alstrom syndrome (Alms) 1*, a causal gene for a rare autosomal recessive disorder characterized by childhood obesity and sensory impairment, categorized as multiorgan ciliopathy (*Marshall et al., 2011*). The mammalian Alms1 protein was previously shown to localize to centrosomes and basal bodies of the primary cilia (*Andersen et al., 2003*; *Hearn et al., 2005*). The

*Drosophila* genome encodes two *Alms1* homologs, juxtaposed to each other on the X chromosome, which we named *alms1a (CG12179)* and *alms1b (CG12184)*. The peptide sequences identified by our mass-spectrometry analysis were identical between these two proteins. We raised specific antibodies against Alms1a and Alms1b, and found that Alms1a localized to centrosomes in all germ cells (GSCs to spermatids) (*Figure 1C* and *Figure 1—figure supplement 1*), whereas Alms1b was not expressed until a late stage of spermatogenesis and localized to the basal body of elongating spermatids (*Figure 1—figure supplement 1D,E*). Therefore, Alms1a is likely the relevant Klp10A interactor in GSCs: accordingly, we focused on Alms1a in this study.

Alms1a and Klp10A co-immunoprecipitated in GSC-enriched extract but not in SG-enriched extract, confirming that Alms1a is a GSC-specific Klp10A interactor (*Figure 1D*). We found that this physical interaction between Alms1a and Klp10A is direct, and mediated via Alms1a's conserved C-terminal region ('ALMS motif'), as demonstrated by pulldown of the GST-fusion protein expressed in bacterial cells (*Figure 1E–F*) as well as by a bacterial two hybrid method (*Figure 1—figure supplement 2*).

We further assessed the interaction between Alms1a and Klp10A using the bimolecular fluorescence complementation (BiFC) method (*Hu et al., 2002*), which reveals protein-protein interactions in vivo via reconstitution of a functional Venus YFP from two non-fluorescent fragments (VNm9 and VC155) (*Saka et al., 2007*). Upon co-expression of *alms1a*-VNm9 and VC155-*klp10A* in the germline (*nos-gal4>UAS-alms1a-VNm9, UAS-VC155-klp10A*), we observed a YFP signal at centrosomes in GSCs, but not in the GB and SGs (*Figure 1G,H* and *Figure 1—figure supplement 3*). The lack of BiFC signals in the GB and SG serves as an internal negative control and confirms that the interaction between Alms1a and Klp10A is GSC-specific. These results imply that, although Alms1a localizes to the centrosomes of both GSCs and SGs, Alms1a might have a specific function in GSCs via its interaction with Klp10A.

## Alms1a localizes asymmetrically between mother versus daughter centrosomes in male GSCs

To begin to understand the function of Alms1a, we first examined its centrosomal localization in GSCs and SGs in more detail. Each centrosome consists of two centrioles, the mother centriole ('m' in *Figure 2A*) and daughter centriole ('d' in *Figure 2A*). Centrosome duplication generates a mother centrosome that contains the original mother centriole ('M' in *Figure 2A*), and a daughter centrosome that contains the original daughter centriole, now a new mother ('D' in *Figure 2A*).

We found that, although Alms1a localized to centrosomes in all cell types, it was more enriched on the mother centrosome than the daughter centrosome in GSCs (*Figure 2B–D,K*). In contrast, the two centrosomes in SGs had an equivalent amount of Alms1a (*Figure 2E–G,K*). Knockdown of *klp10A* in the male germline (*nos-gal4>UAS-klp10A*[TRiP.HMS00920], *UAS-klp10A*[RNAi] hereafter [*Chen et al., 2016*]) abolished the asymmetric enrichment of Alms1a to the mother centrosome in GSCs (*Figure 2H–K*), suggesting that Klp10A is necessary to create the asymmetrical distribution of Alms1a to the mother vs. daughter centrosomes in GSCs. *klp10A*[RNAi] did not affect Alms1a localization to the SG centrosomes (*Figure 2—figure supplement 1A*). Germline knockdown of *alms1a* (*nos-gal4>alms1*[TRiP.HMJ30289], referred as to *alms1*[RNAi] hereafter) did not compromise Klp10A localization in GSCs (*Figure 2—figure supplement 1B,C*).

We further investigated the localization of Alms1a within the centrosome by examining its spatial relationship with known centriolar markers. Alms1a localization to the centrosome mostly did not overlap with Centrobin (Cnb), a daughter centriole marker (*Januschke et al., 2013*; *Figure 2B,D–G*), suggesting that Alms1a might localize to the mother centriole (see below). Detailed examination of Alms1a localization using Lightning confocal microscopy (see Materials and methods) confirmed that Alms1a is a mother centriole protein in GSCs and GBs/SGs (*Figure 2L–N*). Ana1 is a core centriole protein required for centrosome assembly and is known to localize to the mother centriole in early interphase (*Fu et al., 2016*; *Goshima et al., 2007*), whereas Cp110 localizes to the distal end of both mother and daughter centrioles, where it 'caps' the centriole (*Delgehyr et al., 2012*; *Schmidt et al., 2009*). We found that Alms1a localizes to the proximal end of the mother centriole: it localizes to the Ana1-positive centriole (e.g. mother centriole) at the opposite end compared to CP110, which is at the distal end (*Figure 2M,N*). Its localization to the proximal end was clearly visible in the spermatocytes (SCs), where much longer centrioles allowed easier distinction of distal vs. proximal ends (*Figure 2O*).

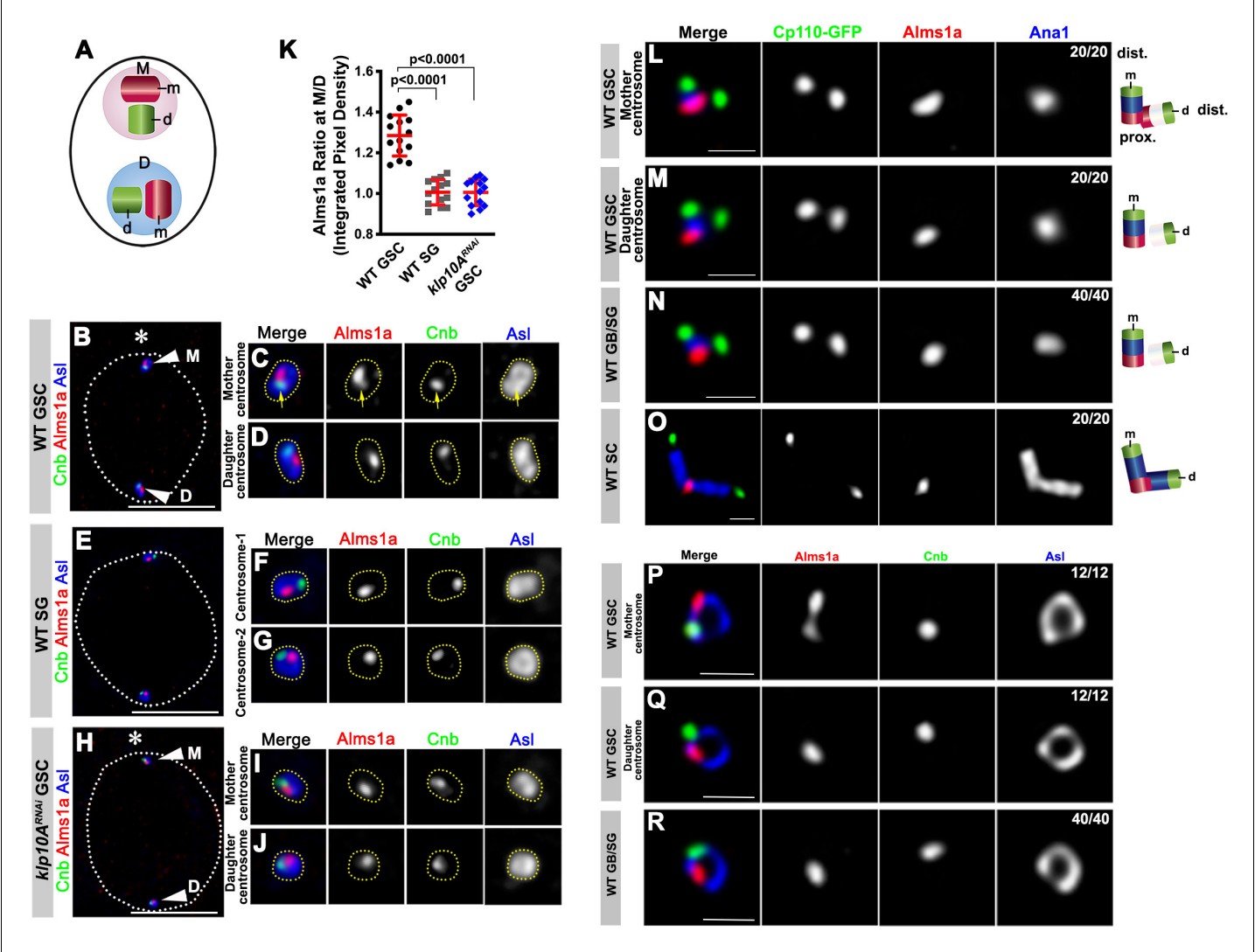

**Figure 2.** Alms1a exhibits asymmetric localization to the mother centrosome in germline stem cells (GSCs). (A) Schematic of centrosome structure. Each cell contains two centrosomes, mother centrosome (M, pink) and daughter centrosome (D, blue). Each centrosome consists of two centrioles: mother centriole (m, dark purple) and daughter centriole (d, light purple). (B–D) A HyVolution image of wild-type GSC centrosomes (B). (C–D) Magnified images of a mother centrosome (C) and a daughter centrosome (D) from panel (B). Green: Centrobin (Cnb). Red: Alms1a. Blue: Asl. Asterisk indicates the hub. Arrowheads indicate GSC centrosomes. Arrow indicates mother centrosome. Bar: 5 μm. (E–G) A HyVolution image of wild type SG centrosomes (E). (F–G) Magnified images of SG centrosomes from panel (E). (H–J) A HyVolution image of *klp10A^RNAi* GSC centrosomes (H). (I–J) Magnified images of a mother centrosome (I) and a daughter centrosome (J) from panel (H). (K) Ratio of integrated pixel density of Alms1a signal between mother and daughter centrosomes (M/D) in the indicated genotypes. P value was calculated using two-tailed Student's t-test. Error bars indicate the standard deviation. 14 cells were scored for each data point. (L–O) Lightning images of a GSC mother centrosome (L), a daughter centrosome (M), a GB/SG centrosome (N) and a SC centrosome (O). Green: Cp110-GFP. Red: Alms1a. Blue: Ana1. Bar: 0.5 μm. Cartoon shows the interpretation of protein localization to the mother and daughter centrioles. (P–R) Stimulated emission depletion (STED) super-resolution images of wild-type GSC mother centrosome (O), a GSC daughter centrosome (P), and a SG centrosome (Q). Red: Alms1a. Blue: Asl. Green: Cnb. Bar: 0.5 μm.

The online version of this article includes the following figure supplement(s) for figure 2:

**Figure supplement 1.** *alms1a* is not required for Klp10A localization to germline stem cell (GSC) centrosomes.

**Figure supplement 2.** Stimulated emission depletion (STED) super-resolution images of endogenous Asl and Cep135 localization throughout the cell cycle in germline stem cells (GSCs).

In addition to this localization at the proximal end of the mother centriole, Alms1a appeared to extend toward the proximal end of the daughter centriole specifically in the GSC mother centrosome (*Figure 2L*). This localization likely accounts for a higher amount of Alms1a at the GSC mother centrosome (*Figure 2B–D,K*). However, the Lightning microscopy did not resolve whether Alms1a at the mother and daughter centrioles are continuous or distinct entities. To achieve better resolution, we used stimulated emission depletion (STED) super-resolution microscopy: the mother centriole was marked by Asl, which is known to surround the mother centriole in a ring shape (*Figure 2—figure supplement 2*). The Asl ring observed in GSC centrosomes matches to the dimension of the Asl ring reported in other *Drosophila* cells (*Fu et al., 2016*; *Galletta et al., 2016*). We found that Alms1a staining appears as two spots within the GSC mother centrosome (*Figure 2P*), but as one spot within the GSC daughter centrosome and GB/SG centrosomes (*Figure 2Q–R*). Together, these results demonstrate that Alms1a localizes to the mother and daughter centrioles at the GSC mother centrosomes, but only to the mother centrioles elsewhere, generating an asymmetry between mother vs. daughter centrosomes of GSCs.

It should be noted that, although Klp10A was reported to be more concentrated at the distal end of the centriole in *Drosophila* spermatocytes (*Delgehyr et al., 2012*; *Riparbelli et al., 2018*), Klp10A exhibits pan centrosomal (broadly pericentrosomal) localization in GSCs (*Chen et al., 2016*). We suggest that Klp10A-Alms1a interaction likely occurs at the proximal end of the centrioles, where Alms1a is most concentrated, although other possibilities as to the site of their functional interaction cannot be excluded.

## Depletion of *alms1a* leads to centrosome loss due to defective centriole duplication

To understand the function of *alms1a*, we examined the phenotypes arising from depletion of *alms1a*. Depletion of *alms1a* was confirmed using a specific antibody against Alms1a (*Figure 1—figure supplement 1A–C and G*). *UAS-alms1$^{RNAi}$* is expected to knockdown both *alms1a* and *alms1b*, due to the high degree of their sequence similarity: However, because *alms1b* is not visibly expressed in GSCs/SGs (*Figure 1—figure supplement 1D*), the phenotypes caused by *alms1$^{RNAi}$* in GSCs are likely due to the loss of *alms1a* (see Materials and methods). Expression of the full-length Alms1a (*nos-gal4>alms1a-GFP,* made to be insensitive to RNAi, see Materials and methods) rescued the centrosome loss phenotype of *alms1$^{RNAi}$* phenotype partially, suggesting that the phenotype is caused, at least in part, by the loss of *alms1a* (*Figure 3—figure supplement 1*). The partial rescue may be due to an insufficient level of transgene expression, as *nos-gal4* often causes variable expression levels among cells. Alternatively, GFP-tagging of Alms1a in the rescue transgene may partially compromise its functionality, although GFP-tagged Alms1a exhibits identical localization pattern as endogenous Alms1a detected by antibody staining. Also, the possibility of off-target effect of RNAi construct cannot be entirely excluded, although it is unlikely, given that the target sequence of *alms1$^{RNAi}$* does not have any other similar sequences within the *Drosophila* genome.

Knockdown of *alms1a* (*nos-gal4>UAS-alms1$^{RNAi}$*, whereby RNAi is induced starting from embryogenesis) resulted in the complete loss of centrosomes from all germ cells, except for the mother centrosomes in GSCs, suggesting that *alms1a* is required for generating new centrosomes/centrioles (*Figure 3A,B*). Wild-type GSCs normally contain mother and daughter centrosomes, each of which contains both mother and daughter centrioles, throughout the cell cycle, due to early duplication of the centrosome right after the previous mitosis (*Figure 3C*). In contrast, many *alms1$^{RNAi}$* GSCs contained only one centrosome (i.e. mother), visualized by Asl as well as γ-Tubulin (*Figure 3D*). Interestingly, the remaining centrosome in *alms1$^{RNAi}$* GSCs gradually elongated (*Figure 3D* and *Figure 3—figure supplement 2*), similar to our earlier observation with *klp10A$^{RNAi}$* (*Chen et al., 2016*): in *klp10A$^{RNAi}$* GSCs, the mother centrosome gradually elongated, whereas the daughter centrosome was slightly smaller than normal, although it was not lost as is observed in *alms1$^{RNAi}$* GSCs. These data suggest that *klp10A* and *alms1a* share a similar function, possibly balancing the growth of mother vs. daughter centrosomes (see Discussion). We conclude that *alms1a* is required for centrosome duplication in GSCs.

To understand the precise steps of daughter centrosome loss upon depletion of *alms1a*, we induced *alms1$^{RNAi}$* in a temporally controlled manner (*nos-gal4ΔVP16, tub-gal80$^{ts}$>UAS-alms1$^{RNAi}$*): *alms1$^{RNAi}$* was induced upon eclosion and the outcome was monitored thereafter. This approach confirmed that GSCs lose their daughter centrosomes as fast as 1–2 days (2–4 GSC cell cycles) upon

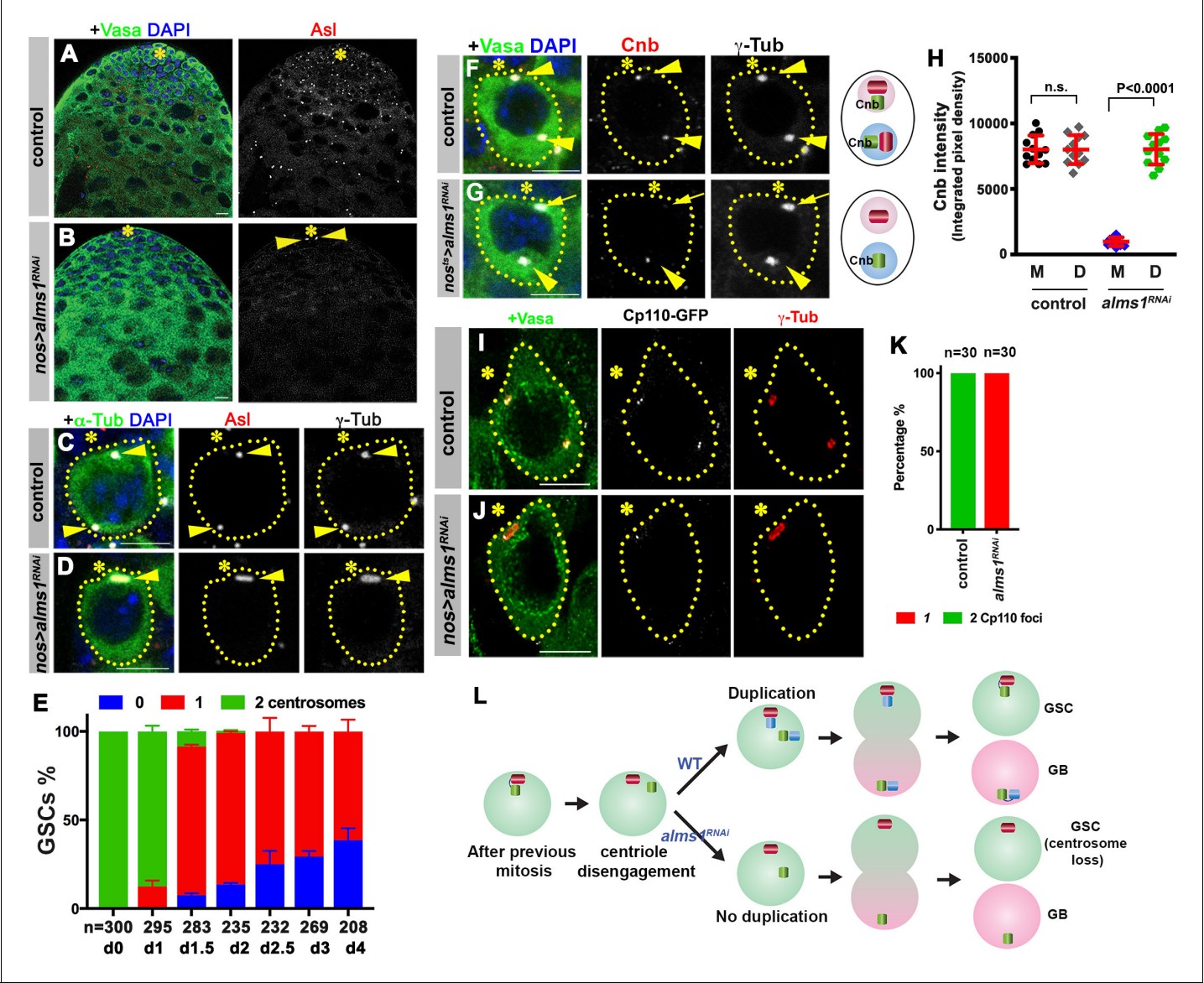

**Figure 3.** *alms1a* is required for daughter centriole duplication in germline stem cells (GSCs). (**A–B**) Examples of apical tips in control (**A**) and *nos-gal4>UAS-alms1^{RNAi}* (**B**) testes (in this genotype, RNAi is induced since embryogenesis). Green: Vasa. Red: Asl. Blue: DAPI. Asterisk indicates the hub. Arrowheads indicate remaining centrosomes in *alms1^{RNAi}* GSCs. Bar: 10 μm. (**C–D**) Examples of centrosomes in control (**C**) and *nos-gal4>UAS-alms1^{RNAi}* (**D**) GSCs. Green: GFP-α-tubulin. Red: Asl. White: γ-Tub. Blue: DAPI. GSCs are outlined with yellow dotted lines. Arrowheads indicate GSC centrosomes. Bar: 5 μm. (**E**) Quantification of centrosome number in temporarily controlled *alms1^{RNAi}* GSCs. (0–4 days after RNAi induction, *nos-gal4ΔVP16, tub-gal80^{ts}>UAS-alms1^{RNAi}*). Centrosome number was quantified by Asl and γ-Tub double staining. Error bars indicate the standard deviation. n = GSC numbers scored for each data point. (**F, G**) Examples of GSCs in control GSC (**F**) and *alms1^{RNAi}* GSC (G, 2 days after RNAi induction, *nos-gal4ΔVP16, tub-gal80^{ts}>UAS-alms1^{RNAi}*) stained for daughter centriole protein Cnb. Green: Vasa. Red: Cnb. White: γ-Tub. Blue: DAPI. Arrowheads indicate GSC centrosomes. Arrow indicates mother centrosome in *alms1^{RNAi}* GSC lacking Cnb staining (2 days after RNAi induction). The daughter centrioles are marked by Cnb. Cartoon shows interpretation of centriole composition. The color scheme is the same as *Figure 2A*. Bar: 5 μm. N = 30 GSCs for each genotype. (**H**) Quantification of Cnb amount (integrated pixel density) on the mother (M) vs. daughter (D) centrosomes in the indicated genotypes. p-Value was calculated using two-tailed Student's t-test. Error bars indicate the standard deviation. Twelve cells were scored for each data point. (**I, J**) Lightning images of GSCs in control (**I**) and *nos-gal4>UAS-alms1^{RNAi}* (**J**) testes. Green: Vasa. Red: Asl. White: Cp110. Asterisk indicates the hub. Bar: 10 μm. (**K**) Quantification of Cp110 dot number/centrosome in control vs. *alms1^{RNAi}* GSCs. (**L**) Model of centrosome loss in *alms1^{RNAi}* GSCs. Red centriole: the original mother centriole. Green centriole: the original daughter centriole that becomes a mother for the first time after centriole duplication. Blue centriole: the newest daughter centriole generated in the latest cell cycle.

The online version of this article includes the following figure supplement(s) for figure 3:

**Figure supplement 1.** *alms1a* transgene partially rescues the centrosome loss phenotype in *alms1^{RNAi}*.

*Figure 3 continued on next page*

*Figure 3 continued*

**Figure supplement 2.** Quantification of germline stem cell (GSC) mother centrosome length after the induction of *alms1$^{RNAi}$*.

**Figure supplement 3.** Fertility of control, *nos-gal4>UAS-alms1$^{RNAi}$* and *bam-gal4>UAS-alms1$^{RNAi}$* males.

RNAi-mediated knockdown of *alms1a* (*Figure 3E*), and allowed us to examine the process of centrosome loss. In control GSCs, the daughter-centriole-specific protein Cnb always marked both mother and daughter centrosomes, as expected because both centrosomes contain a daughter centriole (*Figure 3F,H*). However, upon knockdown of *alms1a*, Cnb is lost or diminished from the mother centrosome prior to centrosome loss: when GSCs still contained two γ-Tubulin-positive centrosomes, the mother centrosome lacked Cnb staining, suggesting that it does not contain the daughter centriole (*Figure 3G,H*). Based on these results, we infer that centriole duplication is compromised in GSCs upon *alms1* depletion: mother centriole (lacking Cnb) and daughter centriole (marked by Cnb) of the GSC mother centrosome likely split from each other, and segregate into GSC and GB, respectively, without duplicating centrioles (*Figure 3L*). In the following cell cycle, the GSCs that are incapable of duplicating centrioles will continue to divide without passing any new centrosomes to GBs. The inability of *alms1$^{RNAi}$* GSCs to duplicate centrioles was further comfirmed by detailed examination of centrosomes upon complete loss of the daughter centrosomes in *alms1$^{RNAi}$* GSCs. In contrast to control GSCs, which contain two centrosomes, each of which contains two centrioles marked by Cp110 (*Figure 3I,K*), *alms1$^{RNAi}$* GSCs contained only one centrosome, which contains only one centriole (*Figure 3J,K*), demonstrating that *alms1$^{RNAi}$* compromises centriole duplication. The lack of centriole duplication in GSCs would lead to loss of centrosomes/centrioles in all germ cells, which must inherit the first template for centriole duplication from GSCs (*Figure 3L*). *nos-gal4>UAS-alms1$^{RNAi}$* flies were sterile, likely due to the complete loss of centrosomes, although knockdown of *alms1b* also contributes to this sterility (*Figure 3—figure supplement 3*). Taken together, we conclude that *alms1a* plays a critical role in ensuring centrosome duplication in GSCs, which is essential for the production of centrosomes/centrioles in all downstream germ cells.

## *alms1a* is dispensable for centrosome duplication in differentiating germ cells and GSCs that are induced to divide symmetrically

The above results show that *alms1a* is required to ensure the duplication of daughter centrioles in GSCs. Considering the unique asymmetric localization of Alms1a only in GSCs but not in differentiating cells, we wondered whether Alms1a is required for centriole duplication in differentiating cells.

To address this question, we depleted *alms1a* from differentiating cells. The SG-specific driver *bam-gal4* was used to deplete *alms1a* in differentiating germ cells (*bam-gal4>UAS-alms1$^{RNAi}$*). We found that SGs, which divide symmetrically, maintained their centrosomes upon depletion of *alms1a* (*Figure 4A*). Thus, *alms1a* is specifically required for centriole duplication in stem cells, but not in symmetrically dividing, differentiating cells. Note that this result was not due to incomplete depletion of *alms1a*, as Alms1a was undetectable in SGs in the testis from *bam-gal4>UAS-alms1$^{RNAi}$* flies (*Figure 4A*).

To test whether the unique requirement of *alms1a* in GSCs is due to their asymmetric division or to their stem cell identity, we examined whether symmetrically dividing GSCs require *alms1a* for centriole duplication (*Figure 4B–C*). We induced symmetric divisions in GSCs by expressing *upd* together with *alms1$^{RNAi}$* (*nos-gal4>UAS-upd, UAS-alms1$^{RNAi}$*). Alms1a protein was undetectable under this condition, confirming efficient knockdown (*Figure 4C*). Importantly, these symmetrically dividing GSCs retained their centrosomes (*Figure 4C–D*), suggesting that *alms1a* is required for centriole duplication only in asymmetrically dividing cells.

## Alms1a interacts with the core centriole duplication machinery

To further understand how Alms1a regulates centriole duplication in the GSCs, we examined whether Alms1a interacts with key proteins for centriole biogenesis. Sak is a homolog of the Plk4 kinase, a master regulator required for centriole duplication (*Bettencourt-Dias et al., 2005*; *Habedanck et al., 2005*), whereas Sas-6 is a component for cartwheel assembly to promote centriole formation (*Nakazawa et al., 2007*; *Rodrigues-Martins et al., 2007a*). Alms1a physically interacted with Sak and Sas-6 in testes lysates enriched with GSCs, as indicated by co-

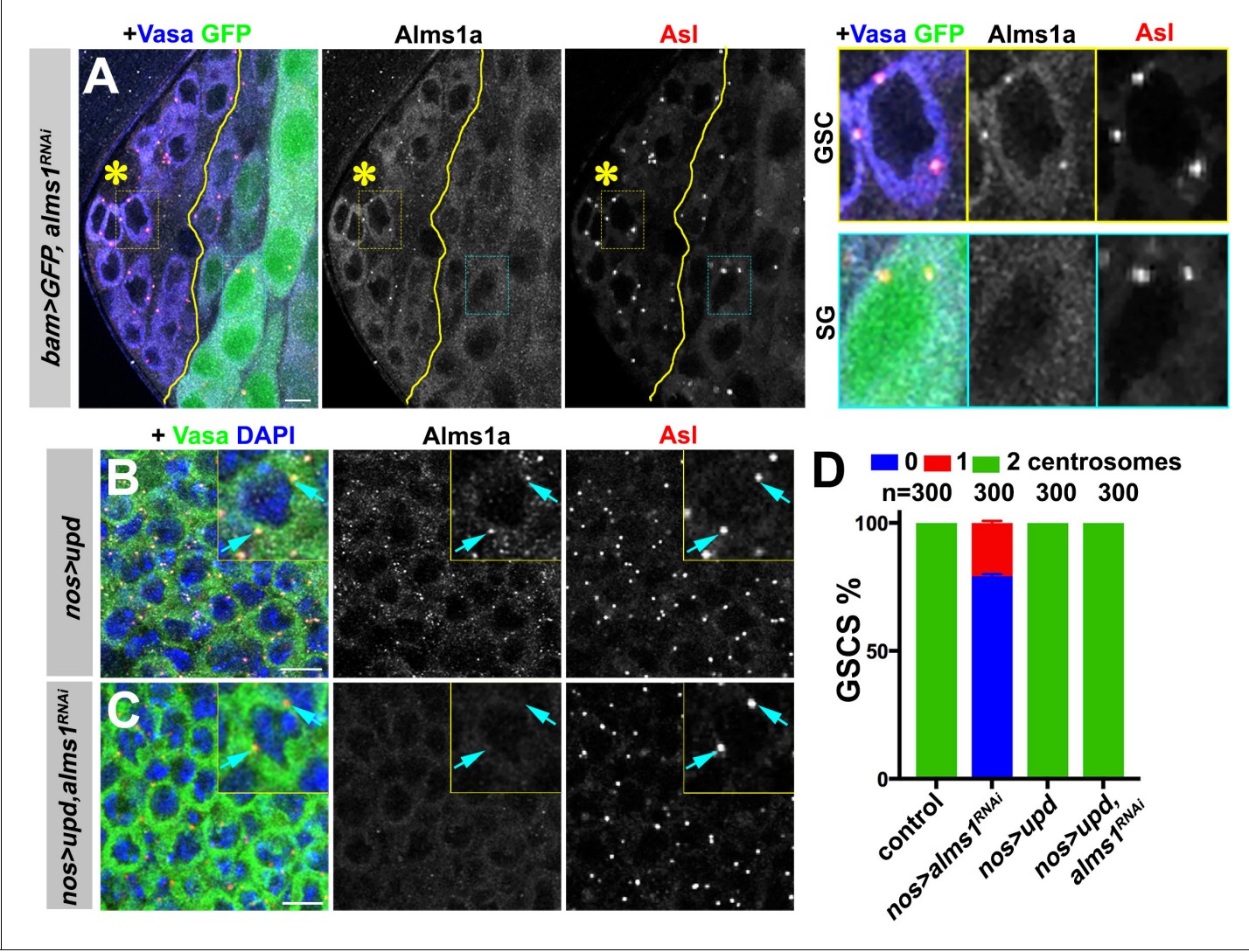

**Figure 4.** *alms1a* is dispensable for centrosome duplication in symmetrically dividing cells. (**A**) An example of Alms1a staining in *bam-gal4>UAS-GFP, UAS-alms1^RNAi* testis. The boundary of bam-gal4-positive vs. -negative SGs is indicated by the yellow line (note that bam expression marked by GFP is initially weak). Green: GFP. White: Alms1a. Red: Asl. Blue: Vasa. Asterisk indicates the hub. Yellow box inset indicates an example of Alms1a staining in germline stem cell (GSC). Cyan box inset indicates an example of Alms1a staining in SG. Bar: 10 μm. (**B–C**) Examples of *upd*-induced GSCs without *alms1^RNAi* (B, *nos-gal4>UAS-upd*) or with *alms1^RNAi* (C, *nos-gal4>UAS-upd, UAS-alms1^RNAi*). Green: Vasa. Red: Asl. White: Alms1a. Blue: DAPI. Inset is magnified image of a single GSC. Arrows indicate centrosomes. Bar: 10 μm. (**D**) Quantification of centrosome numbers (quantified by Asl and γ-Tub double staining) in the indicated genotypes. Error bars indicate the standard deviation. n = GSC numbers scored.

immunoprecipitation (*Figure 5A–B*). Moreover, GST-pulldown experiments using bacterial extracts as well as bacterial two hybrid experiments showed that Sak directly binds to Alms1a via the ALMS motif. In contrast, Sas-6-ALMS motif did not interact in bacterial extract (*Figure 5C–E* and *Figure 5— figure supplement 1*), suggesting that the interaction between Alms1a and Sas-6 in testis extract is indirect.

To further characterize these interactions, we performed the BiFC analysis between Alms1a and Sak. We observed a weak BiFC signal at centrosomes in general, but the BiFC signal was significantly higher at the GSC mother centrosome (*Figure 5F–L*). Given that Alms1a localizes to the mother centriole in all centrosomes, but to the daughter centriole only in GSCs (*Figure 2*), we infer that the Alms1a-Sak interaction preferentially occurs at the daughter centriole of the GSC mother centrosome.

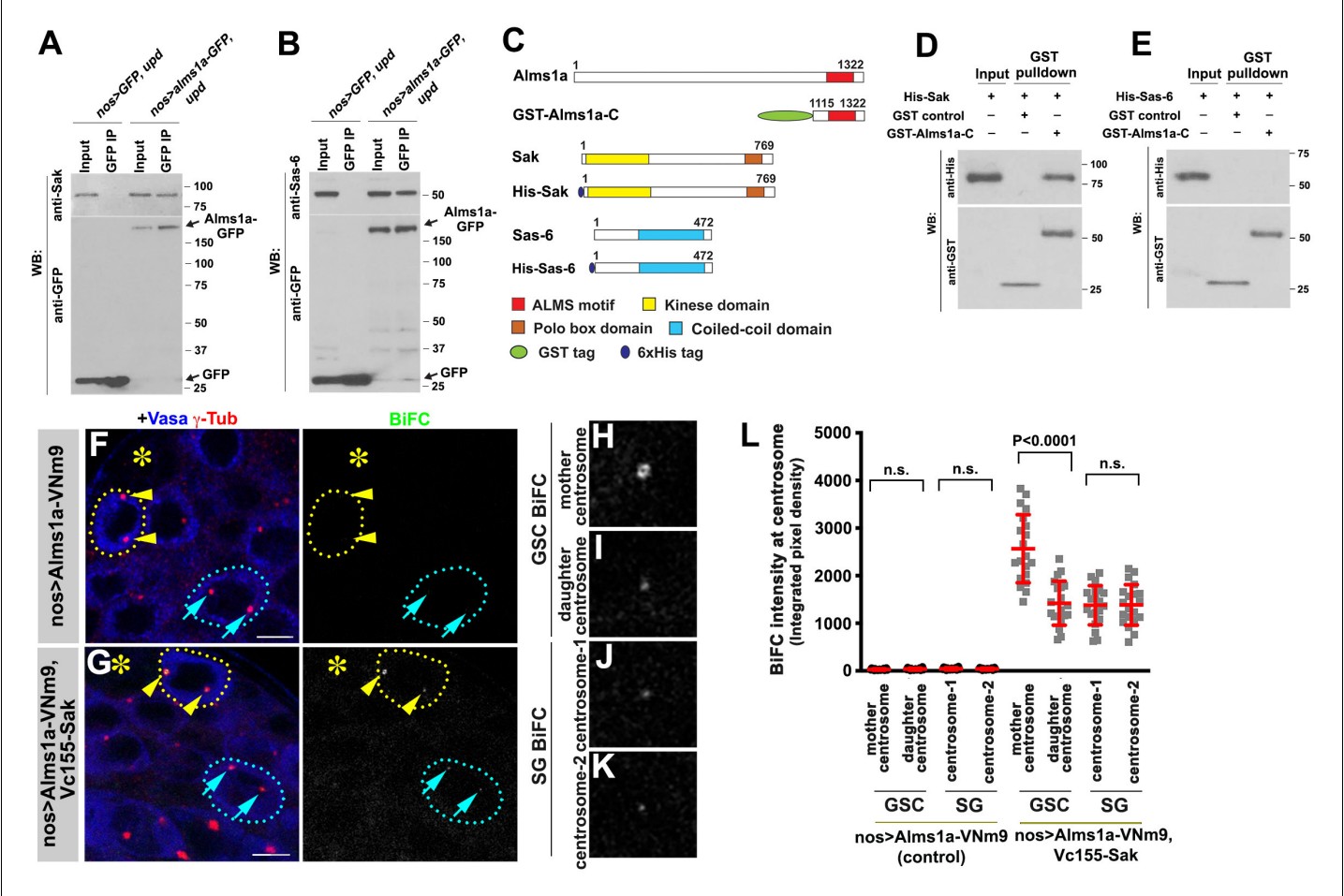

**Figure 5.** Alms1a interacts with Sak and Sas-6 in germline stem cells (GSCs). (A) Co-immunoprecipitation of Sak and Alms1a-GFP were immunoprecipitated from GSC-enriched extracts (*nos-gal4>UAS-upd*) using an anti-GFP antibody and blotted with anti-Sak and anti-GFP. (B) Co-immunoprecipitation of Sas-6 and Alms1a. Control GFP and Alms1a-GFP were immunoprecipitated from GSC-enriched extracts (*nos-gal4>UAS-upd*) using an anti-GFP antibody and blotted with anti-Sas-6 and anti-GFP. (C–H) BiFC analysis of Alms1a-Sak interaction. (C) Schematic of Alms1a, Sak and Sas-6 and constructs used for in vitro pull-down experiments. (D–E) GST-pulldown of Sak, Sas-6 and Alms1a C-terminal fragment. GST control and recombinant GST-Alms1a-C was immobilized on glutathione agarose beads and incubated with His-Sak (D) or His-Sas-6 (E) lysate for pull-down assay and blotted with anti-GST and anti-His antibody. (F–G) Examples of apical tips in *nos-gal4 >UAS-alms1a-VNm9* (control) and *nos-gal4>UAS-alms1a-VNm9, UAS-VC155-sak* testes. Flies are raised at 18°C to minimize centrosome over-duplication due to ectopic protein expression. (H–K) Magnified images of a GSC mother centrosome (H), a GSC daughter centrosome (I) and two SG centrosomes (J–K) from panel (G). Green: BiFC. Red: γ-Tub. Blue: Vasa. Asterisk indicates the hub. GSCs are outlined with yellow dotted lines. Arrowheads (yellow) indicate examples of GSC centrosomes. Arrows (cyan) indicate examples of SG centrosomes. Bar: 5 µm. (L) Quantification of BiFC integrated pixel density in the indicated genotypes. P value was calculated using two-tailed Student's t-test. Error bars indicate the standard deviation. Twenty cells were scored for BiFC signals for each data point.

The online version of this article includes the following figure supplement(s) for figure 5:

**Figure supplement 1.** Bacterial two-hybrid assay showing the interactions between full-length Alms1a and full-length Sak or Sas-6.

## Alms1a promotes centriole duplication together with Sak

How does Alms1a regulate centriole duplication, and why it is required only in GSCs? Alms1a's direct interaction with Sak (*Figure 5*) indicates that Alms1a might promote centriole duplication together with Sak. Indeed, we found that overexpression of Alms1a dramatically enhanced centrosome overduplication caused by Sak overexpression. Overexpression of *alms1a* (*nos-gal4 >UAS-alms1a-GFP, 25°C*) alone did not result in an obvious increase in centrosome number compared to the control (2.09 ± 0.38 centrosomes/cell in *alms1a* overexpression, vs. 2.00 ± 0.00 centrosomes/cell in control) (*Figure 6A,B,E*). Sak overexpression increased centrosome numbers slightly (2.79 ± 1.07

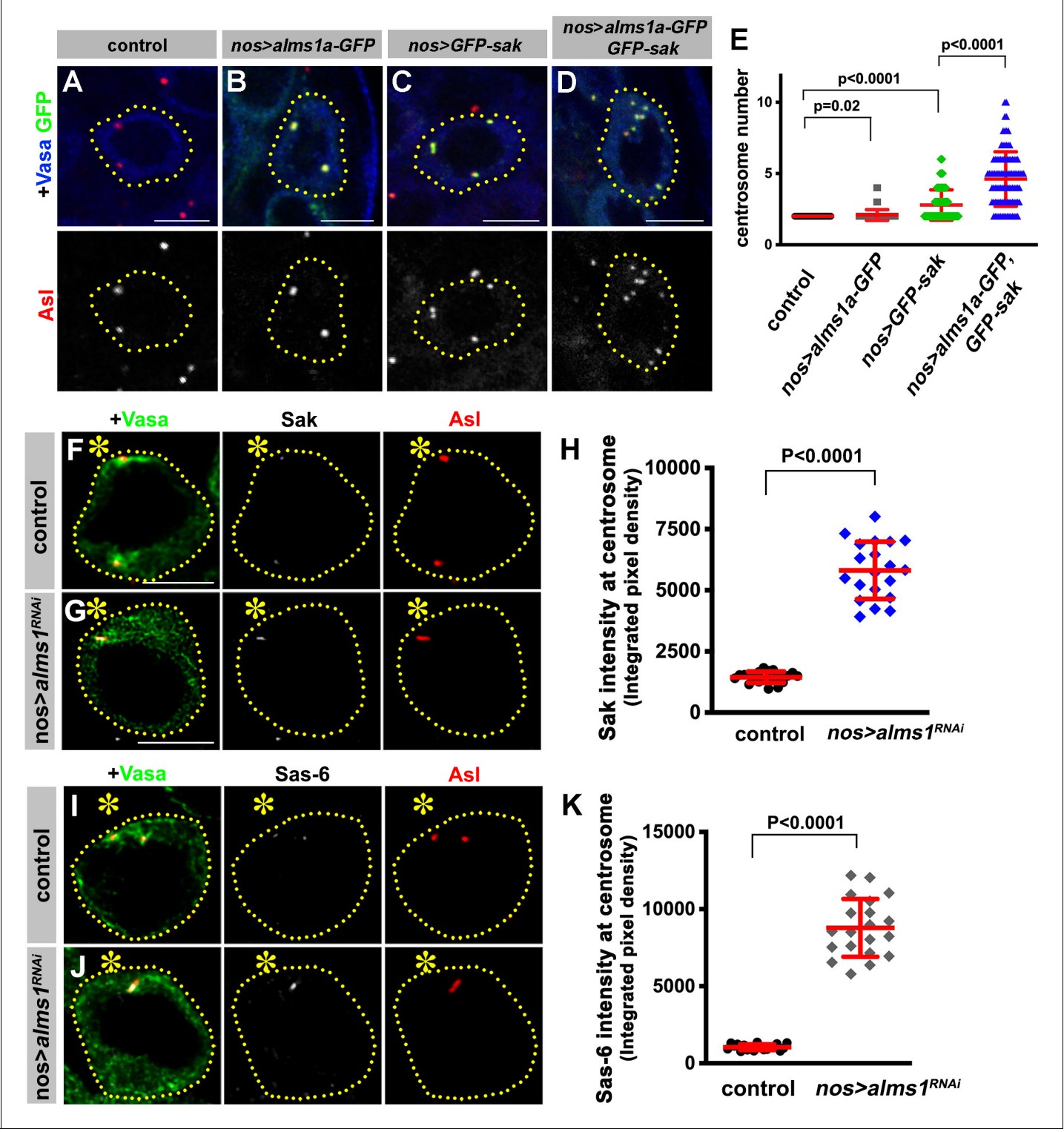

**Figure 6.** Alms1a promotes centrosome duplication together with Sak. (**A–E**) Alms1a overexpression enhances centrosome overduplication induced by Sak overexpression. Examples of germ cells in control (**A**), *nos-gal4>UAS-alms1a-GFP* (**B**), *nos-gal4>UAS-GFP-sak* (**C**) and *nos-gal4>UAS-alms1a-GFP, UAS-GFP-sak* (**D**). Flies were raised at 25°C to drive a higher level of protein expression. Green: GFP. Red: Asl. Blue: Vasa. Germ cells are outlined with yellow dotted lines. Bar: 5 μm. (**E**) Quantification of centrosome number (quantified by Asl and γ-Tub double staining) in GSCs/SGs in indicated genotypes. p-Value was calculated using two-tailed Student's t-test. Error bars indicate the standard deviation. n = 100 cells were scored for each data point. (**F–K**) Sak and Sas-6 accumulate on the remaining mother centrioles in *alms1^RNAi^* GSCs. (**F–G**) Examples of GSC centrosomes in control (**F**) and

*Figure 6 continued on next page*

Figure 6 continued

*nos-gal4>UAS-alms1^RNAi* (G) testes stained for Sak (white), Asl (red), and Vasa (green). Asterisk indicates the hub. GSCs are outlined with yellow dotted lines. Bar: 5 μm. (H) Quantification of Sak integrated pixel density in the indicated genotypes. P value was calculated using two-tailed Student's t-test. Error bars indicate the standard deviation. n = 20 GSCs were scored for Sak signals for each data point. (I–J) Examples of GSC centrosomes in control (I) and *nos-gal4>UAS-alms1^RNAi* (J) testes stained for Sas-6 (white), Asl (red), and Vasa (green). Bar: 5 μm. (K) Quantification of Sas-6 integrated pixel density in the indicated genotypes. P value was calculated using two-tailed Student's t-test. Error bars indicate the standard deviation. n = 20 GSCs were scored for Sas-6 signals for each data point.

The online version of this article includes the following figure supplement(s) for figure 6:

**Figure supplement 1.** Alms1a overexpression enhances centrosome overduplication induced by Sak overexpression.

centrosomes/cell) (*Figure 6C,E*), similar to observations in other cell types (*Habedanck et al., 2005*; *Peel et al., 2007*; *Rodrigues-Martins et al., 2007b*). Strikingly, co-expression of *alms1a* and *sak* (*nos-gal4>UAS-alms1a-GFP, UAS-GFP-sak, 25°C*) dramatically enhanced centrosome overduplication (*Figure 6D–E* and *Figure 6—figure supplement 1*) (4.61 ± 1.91 centrosomes/cell). These results suggest that Alms1a promotes centrosome duplication together with Sak, possibly by recruiting Sak to the site of centriole duplication in GSCs.

Why is Alms1a's function to promote centriole duplication together with Sak only required in GSCs but not in SGs? Interestingly, we found that RNAi-mediated knockdown of *alms1a* in GSCs resulted in the excessive enrichment of both Sak and Sas-6 at the elongated mother centriole that remains after all other centrosomes are depleted (*Figure 6F–K*). The excess amount of Sak and Sas-6 on the remaining mother centriole in *alms1^RNAi* GSCs may explain why it continues to elongate. In parallel, the accumulation of Sak and Sas-6 on the mother centriole may deplete Sak and Sas-6 from the site of the daughter centriole production, leading to the failure in daughter centriole duplication. Therefore, the GSC-specific function of Alms1a may be to promote daughter centriole duplication by recruiting Sak to the 'seeding site' for daughter centriole duplication (see Discussion).

## Discussion

Asymmetric behaviors between mother and daughter centrosomes have been observed in asymmetrically dividing stem cells in several systems (*Conduit and Raff, 2010*; *Habib et al., 2013*; *Januschke et al., 2011*; *Wang et al., 2009*; *Yamashita et al., 2007*), provoking the idea that these centrosome asymmetries may contribute to asymmetric cell fates. Here, we identified a centrosomal protein Alms1a, a homolog of a causative gene of human ciliopathy Alstrom syndrome, that exhibits asymmetric localization between mother and daughter centrosomes in *Drosophila* male GSCs. Alms1a is required for centriole duplication specifically in GSCs, but not in differentiating cells, demonstrating the unique regulation of centrosomes in asymmetrically dividing stem cells.

Although the exact molecular mechanism of Alms1a function remains elusive, a few lines of observations imply its possible function. Alms1a localizes to the mother centriole in all centrosomes, but in GSCs it also localizes to the daughter centriole specifically in the mother centrosome, where it interacts with Sak and Sas-6. Importantly, Alms1a's function is required in GSCs but not SGs, thus this GSC-specific localization at daughter centriole of the mother centrosome is likely what is required for centriole duplication in GSCs. Therefore, we postulate that Alms1a may promote daughter centriole duplication by recruiting Sak (and Sas-6), to the daughter centriole of the GSC mother centrosome.

Why is Alms1a only required in asymmetrically dividing GSCs for centriole duplication? Given that *alms1^RNAi* leads to excessive overgrowth of mother centriole of the GSC mother centrosome at the expense of the daughter centriole duplication, it is tempting to speculate that Alms1a is required to 'discourage' mother centriole's growth and 'encourage' daughter centrioles generation/growth. Indeed, upon *alms1a* depletion, the mother centriole gradually elongates and accumulates excessive amounts of Sak and Sas-6, while failing at daughter centriole duplication. Perhaps to ensure correct centrosome positioning, the mother centriole of the mother centrosome in GSCs may have a stronger microtubule nucleation activity, which may interfere with production of new centrioles. This imbalance between mother strength and daughter duplication may be achieved by Alms1a. Such a scenario would explain why Alms1a is required specifically in asymmetrically dividing GSCs, although the underlying molecular mechanisms on how Alms1a may achieve this remains elusive.

A recent study identified Ninein as a protein that specifically localizes to the daughter centrosome when overexpressed in *Drosophila* neuroblasts and male GSCs (*Zheng et al., 2016*), although no phenotypic consequences were detected upon its depletion, leaving the role of its asymmetric localization unknown. Our Klp10A pull down did not identify Ninein, it will be of interest to study how Klp10A and Alms1a may be functionally related to Ninein.

In summary, our study identified Alms1a as a protein that exhibits asymmetric localization between mother and daughter centrosomes of GSCs. We demonstrate that *alms1a* is uniquely required for centriole duplication in asymmetrically dividing stem cells. This specific requirement of *alms1a* only in a subset of cell types (e.g. asymmetrically dividing adult stem cells) may partly explain the reported late-onset nature of Alstrom syndrome, which exhibits symptoms progressively during childhood, as opposed to many other ciliopathies that are associated with disease symptoms at birth (*Braun and Hildebrandt, 2017*). Our study revealed that GSC centrosomes are indeed under unique regulation, involving GSC centrosome-specific proteins Klp10A and Alms1a. The identification of these proteins (Alms1a, Klp10A [*Chen et al., 2016*] and Ninein [*Zheng et al., 2016*]) paves the way to understand how asymmetric behavior of centrosomes contributes to asymmetric stem cell division.

# Materials and methods

## Key resources table

| Reagent type (species) or resource | Designation | Source or reference | Identifiers | Additional information |
|---|---|---|---|---|
| Strain, strain background (*D. melanogaster*) | *nos-gal4* | PMID:9501989 | | |
| Strain, strain background (*D. melanogaster*) | *bam-gal4* | PMID:12571107 | | |
| Strain, strain background (*D. melanogaster*) | *tub-gal4* | Bloomington Stock Center | ID_BSC: 5138 | |
| Strain, strain background (*D. melanogaster*) | *UAS-upd* | PMID:10346822 | | |
| Strain, strain background (*D. melanogaster*) | *tub-gal80$^{ts}$* | PMID:14657498 | | |
| Strain, strain background (*D. melanogaster*) | *nos-gal4* without VP16 (*nos-gal4ΔVP16*) | PMID:26131929 | | |
| Strain, strain background (*D. melanogaster*) | *UAS-EGFP* | Bloomington Stock Center | ID_BSC: 5430 | |
| Strain, strain background (*D. melanogaster*) | *UAS-dpp* | Bloomington Stock Center | ID_BSC: 1486 | |
| Strain, strain background (*D. melanogaster*) | *yw* | Bloomington Stock Center | ID_BSC: 189 | |
| Strain, strain background (*D. melanogaster*) | *UAS-GFP-alpha-tubulin84B* | Bloomington Stock Center | ID_BSC: 7373 | |
| Strain, strain background (*D. melanogaster*) | *UAS-klp10A$^{RNAi}$* | Bloomington Stock Center | ID_BSC: 33963 | |

*Continued on next page*

*Continued*

| Reagent type (species) or resource | Designation | Source or reference | Identifiers | Additional information |
|---|---|---|---|---|
| Strain, strain background (*D. melanogaster*) | *UAS-klp10A-EGFP* | PMID:26131929 | | |
| Strain, strain background (*D. melanogaster*) | *centrobin-YFP* | PMID:21407209 | | |
| Strain, strain background (*D. melanogaster*) | *ubi-asl-tdTomato* | PMID:21694707 | | |
| Strain, strain background (*D. melanogaster*) | *ubi-ana1-tdTomato* | PMID:18854586 | | |
| Strain, strain background (*D. melanogaster*) | *ubi-Cp110-GFP* | PMID:27185836 | | |
| Strain, strain background (*D. melanogaster*) | *UAS-alms1$^{RNAi}$* | Bloomington Stock Center | ID_BSC: 63721 | |
| Strain, strain background (*D. melanogaster*) | *UAS-alms1a-EGFP* | This study | | |
| Strain, strain background (*D. melanogaster*) | *UAS-alms1a-EGFP* (insensitive to RNAi) | This study | | |
| Strain, strain background (*D. melanogaster*) | *UAS-alms1a-VNm9* | This study | | |
| Strain, strain background (*D. melanogaster*) | *UAS-VC155-klp10A* | This study | | |
| Strain, strain background (*D. melanogaster*) | *UAS-VC155-sak* | This study | | |
| Strain, strain background (*D. melanogaster*) | *UAS-EGFP-ALMS motif* | This study | | |
| Strain, strain background (*D. melanogaster*) | *cep135-EGFP* | Bloomington Stock Center | ID_BSC: 60183 | |
| Antibody | Anti-vasa (Rabbit polyclonal) | Santa Cruz Biotechnology | ID_SCB: d-26 | IF: 1:200 |
| Antibody | Anti-alpha-Tubulin (Mouse monoclonal) | Developmental Studies Hybridoma Bank | | WB: 1:2000 |
| Antibody | Anti-vasa (Rat monoclonal) | Developmental Studies Hybridoma Bank | | IF: 1:20 |
| Antibody | Anti-gamma-Tubulin (Mouse monoclonal) | Sigma-Aldrich | ID_Sigma:T6557 | IF: 1:100 |
| Antibody | Anti-Asl (Rabbit polyclonal) | A gift from Jordan Raff | | IF: 1:5000 |
| Antibody | Anti-Sas-6 (Rabbit polyclonal) | A gift from Jordan Raff | | IF: 1:5000 WB: 1:40000 |

*Continued on next page*

*Continued*

| Reagent type (species) or resource | Designation | Source or reference | Identifiers | Additional information |
|---|---|---|---|---|
| Antibody | Anti-Sak (Rabbit polyclonal) | A gift from Monica Bettencourt-Dias | | IF: 1:400 WB: 1:4000 |
| Antibody | Anti-Spd2 (Rabbit polyclonal) | A gift from Maurizio Gatti | | IF: 1:25 |
| Antibody | Anti-GFP (Chick monoclonal) | Aves Labs | ID_aves: GFP-1020 | WB: 1:6000 |
| Antibody | Anti-GST (Mouse monoclonal) | Biolegend | | WB: 1:4000 |
| Antibody | Anti-His (Mouse monoclonal) | Millipore-Sigma | | WB: 1:4000 |
| Antibody | Anti-Alms1a (Guinea pig polyclonal) | This study | | IF: 1:4000 WB: 1:40000 |
| Antibody | Anti-Alms1b (Guinea pig polyclonal) | This study | | IF: 1:4000 |
| Antibody | Anti-Centrobin (Rabbit polyclonal) | This study | | IF: 1:4000 |
| Recombinant DNA reagent | *pGEX-4T-LP* (plasmid) | Addgene | | |
| Recombinant DNA reagent | *pMCSG7* (plasmid) | Addgene | | |
| Recombinant DNA reagent | *pGEX-alms1a-C* (plasmid) | This study | | |
| Recombinant DNA reagent | *pMCSG7-klp10A* (plasmid) | This study | | |
| Recombinant DNA reagent | *pMCSG7-sak* (plasmid) | This study | | |
| Recombinant DNA reagent | *pMCSG7-sas-6* (plasmid) | This study | | |
| Recombinant DNA reagent | *pKNT-25* (plasmid) | Euromedex | | |
| Recombinant DNA reagent | *pUT18C* (plasmid) | Euromedex | | |
| Recombinant DNA reagent | *pKT25-zip* (plasmid) | Euromedex | | |
| Recombinant DNA reagent | *pUT18C-zip* (plasmid) | Euromedex | | |
| Recombinant DNA reagent | *pKNT-alms1a* (plasmid) | This study | | |
| Recombinant DNA reagent | *pUT-klp10A* (plasmid) | This study | | |
| Recombinant DNA reagent | *pUT-sak* (plasmid) | This study | | |
| Recombinant DNA reagent | *pUT-sas-6* (plasmid) | This study | | |

## Fly husbandry, strains and transgenic flies

All fly stocks were raised on standard Bloomington medium at 25℃, and young flies (0- to 1-day-old adults) were used for all experiments unless otherwise noted. The following fly stocks were used: *nos-gal4* (*Van Doren et al., 1998*), *UAS-upd* (*Zeidler et al., 1999*), *UAS-dpp* (Stock #1486, Bloomington *Drosophila* Stock Center (BDSC)) (*Staehling-Hampton et al., 1994*), *tub-gal80^ts* (Stock # 7017, BDSC) (*McGuire et al., 2003*), *UAS-EGFP* (Stock #5430, BDSC), *yw* (Stock #189, BDSC), *cep135-EGFP* (Stock#60183, BDSC) *bam-gal4* (*Chen and McKearin, 2003*), *tub-gal4* (Stock # 5138, BDSC) (*Lee and Luo, 1999*), *UAS-klp10A-GFP* (*Inaba et al., 2015*), *UAS-GFP-α-tubulin84B* (Stock #7373, BDSC) (*Grieder et al., 2000*), *UAS-klp10A^RNAi* (TRiP.HMS00920, BDSC), *UAS-alms1^RNAi* (TRiP.HMJ30289, BDSC), *nos-gal4* without VP16 (*nos-gal4ΔVP16*) (*Inaba et al., 2015*), *centrobin-YFP* (*Januschke et al., 2011*) (a gift from Cayetano Gonzalez), *ubi-asl-tdTomato* (*Gopalakrishnan et al., 2011*) and *ubi-ana1-tdTomato* (*Blachon et al., 2008*) (gifts from Tomer Avidor-Reiss), *ubi-Cp110-GFP* (*Galletta et al., 2016*) (a gift from Nasser Rusan). Note that *UAS-alms1^RNAi* is expected to knockdown both *alms1a* and *alms1b*. However, because *alms1b* is not expressed in GSCs, the phenotypes resulting from expression of this RNAi construct in GSCs are due to the loss of *alms1a*. Although we attempted to generate *alms1a*-specific RNAi lines, none of them efficiently knocked down *alms1a* as evidenced by western blotting.

For the construction of *UAS-alms1a-VNm9*, the VNm9 fragment was PCR-amplified from a plasmid with pUASP-VNm9 (a gift from Mayu Inaba), and subcloned into NotI/KpnI sites of pUAST-attB. The *alms1a* fragment was PCR-amplified from a cDNA ((LD15034) *Drosophila* Genomics Resource center), and subcloned into BgllI/NotI sites of pUAST-VNm9-attB. For the construction of *UAS-VC155-klp10A and UAS-VC155-sak*, the VC155 fragment was PCR-amplified from a plasmid with pUASP-VC155 (a gift from Mayu Inaba), and subcloned into EcoRI/XhoI sites of pUAST-attB. The *klp10A* fragment was PCR-amplified from a cDNA ((LD29208) *Drosophila* Genomics Resource center), and subcloned into XhoI/KpnI sites of pUAST-VC155-attB. The *sak* fragment was PCR-amplified from a cDNA ((RE70136) *Drosophila* Genomics Resource center), and subcloned into XhoI/KpnI sites of pUAST-VC155-attB. For the construction of *UAS-alms1a-GFP*, *UAS-GFP-ALMS motif* and *UAS-GFP-sak*, the *alms1a*, *ALMS motif* and *sak* fragments were PCR-amplified from cDNAs, and subcloned into BgllI/NotI sites of pUAST-GFP-attB (C-terminal GFP tag) or NotI/KpnI sites of pUAST-GFP-attB (N-terminal GFP tag). *UAS-alms1a-GFP* that is insensitive to RNAi was generated using QuickChange II Site-Directed Mutagenesis Kit (Agilent Technologies) from *UAS-alms1a-GFP* with the following primers:5'-gaattatagtgtattcattcggtttagtctgggtaacctcggatttcggtgatgtgaactgactgatg-3' and 5'-catcagtcagttcacatcaccgaaatccgaggttacccagactaaaccgaatgaatacactataattc-3'. Transgenic flies were generated using PhiC31 integrase-mediated transgenesis at the P{CarryP}attP40 (FBti0114379) or P{CarryP}attP2 (FBst0008622) integration sites (BestGene).

Recombinant protein expression and GST pull-down assays cDNAs for *alms1a*, *klp10A*, *sak* and *sas-6* were purchased from Drosophila Genomics Resource center as described above. Full-length or fragments of these genes were generated by PCR amplification of the cDNAs. For the construction of *pGEX-alms1a-C*, the *alms1a-C* fragment was PCR-amplified from cDNA, and subcloned into BamHI/NotI sites of pGEX-4T-LP (Addgene). Glutathione *S*-transferase (GST)-tagged Alms1a-C (1115-1322aa) or GST alone as a control were expressed in Rosetta (DE3) competent cells (Novagen). For the construction of pMCSG7-*klp10A*, pMCSG7-*sak* and pMCSG7-*sas-6*, full-length genes were amplified from cDNAs, and subcloned into KpnI/EcoRI, KpnI/NotI and KpnI/BamHI sites of pMCSG7 (Addgene), respectively. N-terminal 6xHis-tagged *klp10A, sak* and *sas-6* were expressed in Rosetta competent cells as described above. Cells were grown at 37℃ in Terrific Broth (TB) (Millipore) in the presence of ampicillin (100 μg/mL). After optical density (OD) values reaches to 0.6, protein expression was induced by isopropyl β-D-1-thiogalactopyranoside at a final concentration of 0.3 mM for an additional 20 hr at 16℃. The cells were harvested by centrifugation at 5000 rpm for 5 min, and lysed by sonication in lysis buffer (20 mM Tris-HCl (pH 7.4), 150 mM NaCl, and 1 mM EDTA, 1% NP-40 and 5% glycerol) supplemented with complete protease inhibitor cocktail (Roche) and 1 mM phenylmethyl sulphonyl fluoride (PMSF). The immunoprecipitations between Alms1a-C and Klp10A/Sak/Sas-6 were performed by in vitro GST pull-down assay using Pierce GST Protein Interaction Pull-Down kit (Thermo Scientific). The cell lysates were incubated for 30 min at 4℃ followed by centrifugation at 14,000 rpm for 10 min at 4℃. GST and GST-Alms1a-C lysates with equal total protein level were incubated with glutathione agarose beads for 1 hr at 4℃, and washed three

times in lysis buffer. Then, His-Klp10A, His-Sak and His-Sas-6 lysates with equal total protein level were incubated with washed beads for 2 hr at 4°C, and washed five times in lysis buffer. Before the last wash, the beads were moved to a fresh tube, and eluted with 2xLaemmli buffer (Bio-Rad) containing 5% β-mercaptoethanol, and boiled for 10 min. SDS-PAGE and Western blot were then conducted to examine the protein interactions.

## Bacterial two-hybrid analysis

The bacterial two-hybrid analysis was performed to detect direct protein-protein interactions using bacterial adenylate cyclase-based two-hybrid, BACTH systems (Euromedex). For the construction of *pKNT-alms1a*, the *alms1a* full-length was PCR-amplified from cDNA, and subcloned into HindIII/SmaI sites of *pKNT*25. For the construction of *pUT-klp10A*, *pUT-sak* and *pUT-sas-6*, full-length genes were amplified from cDNAs, and subcloned into KpnI/EcoRI, PstI/KpnI and KpnI/EcoRI sites of *pUT18C*, respectively. Two recombinant plasmids encoding *alms1a-T25* and *T18-klp10A/T18-sak/T18-sas-6* are co-transformed into BTH101 competent cells. Transformants are plated on MacConkey selective plate in the presence of maltose, ampicillin (100 μg/mL) and kanamycin (50 μg/mL), and incubated at 30°C for 24 hr. Complementation can be detected as red colonies on the MacConkey-maltose (Sigma) plates, while colonies will be colorless if no interaction occurs. Co-transformation of pKT25-zip and pUT18C-zip with BTH101 competent cells was used as a positive control. Co-transformation of pKNT25 or pUT18C with one of the recombinant plasmids served as negative controls.

## Immunofluorescent staining and confocal microscopy

*Drosophila* testes were dissected in phosphate-buffered saline (PBS), transferred to 4% formaldehyde in PBS and fixed for 30 min. The testes were then washed in PBST (PBS containing 0.1% Triton X-100) for at least 30 min, followed by incubation with primary antibody in 3% bovine serum albumin (BSA) in PBST at 4°C overnight. Samples were washed for 60 min (three 20 min washes) in PBST, incubated with secondary antibody in 3% BSA in PBST at 4°C overnight, washed as above, and mounted in VECTASHIELD with 4',6-diamidino-2-phenylindole (DAPI; Vector Labs).

The primary antibodies used were as follows: rat anti-Vasa (1:20; developed by A. Spradling and D. Williams, obtained from Developmental Studies Hybridoma Bank (DSHB)); mouse anti-γ-Tubulin (GTU-88; 1:100; Sigma-Aldrich); rabbit anti-Vasa (d-26; 1:200; Santa Cruz Biotechnology); rabbit anti-Asl (1:2000; a gift from Jordan Raff); rabbit anti-Sas-6 (1:2000; a gift from Jordan Raff); rabbit anti-Sak (1:200; a gift from Monica Bettencourt-Dias); rabbit anti-Spd2 (1:25; a gift from M. Gatti); chicken anti-GFP (1:1000, Aves Labs). Anti-Alms1a and Alms1b antibody were generated by injecting peptides (QEMEVEPKKQLEKEQHQNDMQQGEPKGREC) and (CNISQRGNHLEKIE) into guinea pigs (Covance). Affinity purification was used to purify the antibodies. Specificity of the antibodies was validated by the lack of staining in *alms1^{RNAi}* testis (*Figure 1—figure supplement 1*). Anti-Centrobin antibody was generated by injecting a peptide (GRPSRELHGMVHSTPKSGSVEPLRHRPLDDNIC) into rabbits (Covance). Alexa Fluor-conjugated secondary antibodies (Life Technologies) were used with a dilution of 1:200. Images were taken using an upright Leica TCS SP8 confocal microscope with a 63 × oil immersion objective (NA = 1.4) and processed using Adobe Photoshop software. As necessary, better resolution was obtained with HyVolution/Lightning on TCS SP8 confocal microscopy or STED imaging.

## Co-immunoprecipitation and western blotting

For immunoprecipitation using *Drosophila* testis lysate, testes enriched with GSCs due to ectopic Upd expression (100 pairs/sample) were dissected into PBS at room temperature within 30 min. Testes were then homogenized and solubilized with lysis buffer (10 mM Tris-HCl pH 7.5; 150 mM NaCl; 0.5 mM EDTA supplemented with 0.5% NP40 and protease inhibitor cocktail (EDTA-free, Roche)) for 30 min at 4°C. Testes lysates were centrifuged at 13,000 rpm for 15 min at 4°C using a table centrifuge, and the supernatants were incubated with GFP-Trap magnetic agarose beads (ChromoTek) for 4 hr at 4°C. The beads were washed three times with wash buffer (10 mM Tris-HCl pH 7.5; 150 mM NaCl; 0.5 mM EDTA). Bound proteins were resolved in SDS-PAGE and analyzed by western blotting.

For anti-Klp10A pull-down and mass spectrometry assays, testes enriched with GSCs (*nos-gal4>UAS-upd*) or enriched with SGs (*nos-gal4>UAS-dpp*) (250 pairs/sample) were dissected into PBS. Testes were then homogenized and solubilized with lysis buffer (10 mM Tris-HCl pH 7.5; 150

mM NaCl; 0.5 mM EDTA, 0.1% NP40) and protease inhibitor cocktail for 30 min at 4°C. After centrifugation at 13,000 rpm for 15 min at 4°C, supernatants were incubated with anti-Klp10A conjugated Dynabeads for 2 hr at 4°C. The beads were washed three times with wash buffer (10 mM Tris-HCl pH 7.5; 150 mM NaCl; 0.5 mM EDTA), followed by boiling in 2xLaemmli buffer and sent for Mass spectrometry analysis (MS Bioworks).

For western blotting, samples subjected to SDS-PAGE (NuPAGE Bis-Tris gels (8%; Invitrogen)) were transferred onto polyvinylidene fluoride (PVDF) membranes (Immobilon-P; Millipore). Membranes were blocked in PBS containing 5% nonfat milk and 0.1% Tween-20, followed by incubation with primary antibodies diluted in PBS containing 5% nonfat milk and 0.1% Tween-20. Membranes were washed with PBS containing 5% nonfat milk and 0.1% Tween-20, followed by incubation with secondary antibody. After washing with PBS, detection was performed using an enhanced chemiluminescence system (Amersham). Primary antibodies used were rabbit anti-GFP (abcam; 1:4000), guinea pig anti-Alms1a (this study; 1:5000), rabbit anti-Sas-6 (1:10000; a gift from Jordan Raff), rabbit anti-Sak (1:2000; a gift from Monica Bettencourt-Dias), mouse anti-GST (Biolegend; 1:1500), and mouse anti-His (Millipore-Sigma; 1:2500). Secondary antibodies used were goat anti-guinea pig IgG, goat anti-rabbit IgG and goat anti-mouse IgG conjugated with horseradish peroxidase (HRP) (abcam; 1:5000).

## Data analyses

Statistical analysis was performed using GraphPad Prism seven software. Data are shown as means ± standard deviation. The p-value (two-tailed Student's *t-test*) is provided for comparison with the control. Integrated pixel intensity was used to measure the amount of proteins at mother and daughter centrosomes. Regions of interest were manually drawn in ImageJ and background intensity was subtracted from each value.

## Fertility assays

Individual males of the genotypes shown were allowed to mate with two *yw* virgin female flies for 1 week. The flies were removed after a week, and the numbers of progenies were scored.

## Acknowledgements

We thank Drs. Cheng-Yu Lee, Jordan Raff, Cayetano Gonzales, Monica Bettencourt-Dias, Nasser Rusan, Tomer Avidor-Reiss, Mayu Inaba, the Bloomington Stock Center, the Vienna *Drosophila* RNAi Center, the *Drosophila* Genomics Resource Center and the Developmental Studies Hybridoma Bank for reagents, and the Yamashita laboratory, Clemens Cabernard, Sue Hammoud and Life Science Editors for discussions and comments on the manuscript, MS Bioworks for mass-spectrometry analysis. This work was supported by R01GM118308 (to YMY). The research in the Yamashita laboratory is supported by Howard Hughes Medical Institute.

## Additional information

### Competing interests

Yukiko M Yamashita: Reviewing editor, *eLife*. The other author declares that no competing interests exist.

### Funding

| Funder | Grant reference number | Author |
| --- | --- | --- |
| Howard Hughes Medical Institute | | Yukiko M Yamashita |
| National Institute of General Medical Sciences | R01GM118308 | Yukiko M Yamashita |

The funders had no role in study design, data collection and interpretation, or the decision to submit the work for publication.

## Author contributions
Cuie Chen, Conceptualization, Resources, Formal analysis, Investigation, Methodology, Writing - original draft, Writing - review and editing; Yukiko M Yamashita, Conceptualization, Supervision, Funding acquisition, Investigation, Writing - original draft, Writing - review and editing

## Author ORCIDs
Cuie Chen (iD) http://orcid.org/0000-0002-5498-9753
Yukiko M Yamashita (iD) https://orcid.org/0000-0001-5541-0216

## Decision letter and Author response
Decision letter https://doi.org/10.7554/eLife.59368.sa1
Author response https://doi.org/10.7554/eLife.59368.sa2

## Additional files

### Supplementary files
• Supplementary file 1. List of Klp10A-interaction proteins enriched in GSCs identified by mass spectrometry.

• Transparent reporting form

### Data availability
All data generated or analysed during this study are included in the manuscript and supporting files.

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
