## [Decision Letter]

**Acceptance summary:**

This study provides evidence for the role of the Alms1a, a *Drosophila* homolog of the human ciliopathy gene Alstrom syndrome, as a specific centriole duplication factor in male germline stem cells in flies. The study also describes the interactions of Alms1a with some key components of the centriole duplication pathway, and these data will inform further mechanistic studies on the centriole duplication in stem cells and the function of Alms1a in this process.

**Decision letter after peer review:**

Thank you for submitting your article "Alstrom syndrome gene is a stem cell-specific regulator of centriole duplication in the *Drosophila* testis" for consideration by *eLife*. Your article has been reviewed by three peer reviewers, and the evaluation has been overseen by a Reviewing Editor and Anna Akhmanova as the Senior Editor. The following individuals involved in review of your submission have agreed to reveal their identity: Moe R Mahjoub (Reviewer #1); Jens Januschke (Reviewer #2).

The reviewers have discussed the reviews with one another and the Reviewing Editor has drafted this decision to help you prepare a revised submission.

This study investigates the role of Alms1a in regulating the asymmetric behavior of centrosomes in asymmetrically dividing *Drosophila* stem cells. Using a biochemical approach, they identify Alms1a as an interactor of Klp10A, a protein they previously showed to be enriched on centrosomes of male germline stem cells (GSC). They show that Alms1a exhibits asymmetric localization between mother and daughter centrosomes in GSCs, with enrichment on the mother centrosome compared to the daughter centrosome. Using super-resolution imaging, they show that Alms1a localizes to the proximal region of the mother centriole and extends toward the proximal end of the daughter centriole specifically in the GSC mother centrosome. Depletion of Alms1a in vivo during stem cell proliferation resulted in centriole duplication defects, which was specifically associated with asymmetrically dividing GSCs, resulting in loss of centrosomes over time/cell division. Finally, using a combination of biochemical and fluorescence microscopy methods (e.g. BiFC), they demonstrate that Alms1a interacts with key components of the centriole duplication pathway, including Sak and Sas6, to regulate centriole assembly in these stem cells.

The manuscript contains a lot of interesting data and valid points. The discovery of a potential cell type specific centriole duplication factor is very interesting, of broad interest to the readership of *eLife* and thus well worth reporting. The story is consistent with prior results from this lab (and others) that highlight asymmetric behaviors between mother and daughter centrosomes in asymmetrically-dividing stem cells across several systems. On the other hand, mechanistic insights and their interpretation as to how Alms1 would execute this function are as is not straightforward. The proposed model that Alsm1a is a male GSC specific factor that specifically regulates daughter centriole duplication by being present on the daughter centriole within the mother centrosome needs additional support and could benefit, if possible, from more super-resolution data.

Essential revisions:

1) The main observation is based on one RNAi line causing the phenotype. It would be important to add additional controls, a rescue or perhaps the use of a different RNAi line, or a mutant. In addition to the rescue with full-length protein, the authors can attempt the rescue experiment using the C-terminal fragment of the gene, since they suggest this is the key motif sufficient for interactions with both Klp10 at GSC centrosome, and for "recruiting" Sak during centriole duplication. Both points could be addressed with this one approach.

2) Furthermore, while the central observation is remarkable, the study is somewhat preliminary in terms of the molecular mechanism. Refining the localization of Alms1a (and its C-terminal fragment) using a quantitative approach (potentially via STED or another superresolution imaging modality) would also strengthen the conclusion about the relative localization at mother vs daughter centrioles. At the same time, they can use this method to check the status of mother vs daughter centrioles in Alms1a-depleted cells.

3) Figure 6, the overexpression of Alms1 and Sak give rise to extra centrosomes even if Sak alone only results in a milder effect. This is very interesting as Sak over-expression is normally sufficient to generate extra centrosomes. However, the centrosomes in the panel Sak, Alms1 (Figure 6D) have different sizes and Asl intensities. Are the authors sure that they all represent extra bona fide centrosomes? Maybe using other centriole markers, such as Sas4, would be useful.

4) Finally, additional quantifications and controls (e.g. for the bipartide system to detect protein interactions) indicated in specific comments of the reviewers below would be important.

We include below specific comments of individual reviewers for more background.

Reviewer 1:

1) The authors say "RNAi-mediated depletion of klp10A leads to abnormal elongation of the mother centrosomes in GSCs". Do they mean mother centriole? Or are both mother and daughter centrioles of the mother centrosome elongated? This also applies to Figure 3C-D, where the authors suggest "elongated mother centrosomes" upon Alms1 depletion. Based on the centriole duplication defect, I imagine they are referring to the existing mother centriole as being elongated.

2) Figure 1E-F: The authors demonstrate interaction between Alms1a C-terminal motif with Klp10a, indicating this domain is sufficient. Did they test whether the N-terminus plays a role in this too? There are a number of centrosomal proteins that can interact via multiple domains.

3) Figure 2: Is the C-terminal domain of Alms1a, if expressed in the GSC, still show asymmetric association between mother and daughter centrosomes?

4) Figure 2—figure supplement 1A: The authors show that klp10A-RNAi did not affect Alms1a localization to the SG centrosomes. Did they quantify the relative signal intensity between mother and daughter (since depletion of Klp10A does not cause mislocalization of Alms1a from centrosomes in GSCs completely either)?

5) Figure 5: Similar to Figure 1, they show interaction between Alms1a C-terminal motif with Sak, but not with Sas6. Did they identify interaction with Sas6 via the N-terminal half of Alms1a?

6) Figure 6: The authors say "These results suggest that Alms1a promotes centrosome duplication together with Sak, possibly by recruiting Sak to the site of centriole duplication in GSCs". Do the authors know whether Alm1a is phosphorylated, and if so, is this Sak-dependent (since Sak's role in centriole duplication is typically mediated via phosphorylation)?

7) Figure 6: Importantly, if Alms1a's role in duplication is merely to recruit Sak, then one experiment to test that would be to express the C-terminal domain of Alms1a (which they show to be sufficient for interaction), to see if it can rescue the centriole duplication defect in Alms1a-depleted cells.

Reviewer 2:

1) In S-phase both the mother and daughter centrosomes of GSCs contain normally both a set of two centrioles. When centriole duplication is triggered again, both centrioles within the mother centrosome should be competent to duplicate. The data supporting that Alms1a regulates duplication of a specific type of centrioles are currently not strong. Cnb depletion in mother centrioles in GSCs is not quantified. Given that cnb mutant germ line cells do have centrioles despite basal body length regulation problems in spermatocytes (Reina, JCB 2018), it is not clear what depleted Cnb in the Alms1 RNAi GSCs means. While a possibility, it does not demonstrate per se that centriole duplication has not occurred. There is further no solid evidence to support that the mother centriole within the mother centrosome keeps duplicating when Alms1a is depleted, while the daughter centriole does not (appears also unlikely given the morphological changes (elongation) detected. The results point out that Alsm1a localizes to the proximal end of the mother centriole and does not overlap with Cnb, yet the discussion summarizes "Importantly, Alms1a's function is required in GSCs but not SGs, thus this GSC-specific localization at daughter centriole of the mother centrosome is likely what is required for centriole duplication." I got confused here and think that the rationale here needs to be better worked out.

2) Within the GSC centrosomes, would two rings of Asl in GSCs be expected normally in S-phase? This is what Figure 2C, D suggest. However, one Asl ring seems to be detected in Figure 2F, G (SG cells) and in Figure 2O-Q (GSCs). It is possible that Asl labels the wall of only one centriole, despite two centrioles being present as at least in embryos Asl is only incorporated in the new daughter later in the centriole duplication cycle (e.g. Raff lab). Thorough quantifications of the localization of Asl signal (one or two centriole walls detected) is missing. This might be of relevance to assign centriole specific roles, especially as statements on the precise point of the cell and the centriole duplication cycle are tough to make in the system and could potentially be a confounding factor. Is there any way to control that rule that out? It is a pity that the STED analysis had not been extended to measure centrioles upon Alms1a depletion. STED on Asl, Alsm1a and Cnb in controls and upon Alms1 depletion in GSCs plus quantification might be desirable to better understand how loss of a specific pool/enrichment of Alms1 might drive a particular function at a given location.

3) Faulty centriole duplication is a likely possibility of centriole loss here, but centrioles could also be destabilized (fragmented and then lost). Can that be excluded? Especially since the gal80ts experiment revealed that after 4 days almost half of the GSCs appear to be centrosome less (Figure 3H model points at loss of centrosomes from GSCs), while in the nosGa4 driven depletion the conclusion is the GSCs specifically retain centrosomes (quantification missing). Or is mother centrosome anchoring depending on Alms1a? In my understanding the mother centrosome is "immortal" when properly anchored in this system.

Reviewer 3:

1) Alms1 RNAi. The authors show indeed a clear reduction by western blot. Did they rescue the depletion? Even if Alms RNAi results in lower levels, they should show that the phenotypes obtained are only due to this depletion. Using a resistant line or a mutation that can be rescued by Alms1 transgene will show this.

2) Figure 2L, not sure I agree with the interpretation of Alms1 being localized to the proximal site of the two centrioles in the mother centrosome. Maybe super resolution will help and drawing line scans to illustrate the fluorescence intensity distribution?

3) Figure 5E, I am not sure I am interpreting this western blot correctly. While I can see in the IP for Sak the pull down of GST- Alms1 (Figure 5D), in the Sas-6 lane, this is not the case.

4) The bipartide system to detect interactions- VNm9. I wonder if all the controls are included in the quantifications. Because in Figure 5L the interaction in mother GSCs is more prominent, but still is not zero in the other centrosomes. Does it mean that they do not interact or that they interact with more molecules in the mother centrosome?

[Editors' note: further revisions were suggested prior to acceptance, as described below.]

Thank you for submitting your article "Alstrom syndrome gene is a stem cell-specific regulator of centriole duplication in the *Drosophila* testis" for consideration by *eLife*. Your article has been reviewed by three peer reviewers, and the evaluation has been overseen by a Reviewing Editor and Anna Akhmanova as the Senior Editor. The following individuals involved in review of your submission have agreed to reveal their identity: Moe R Mahjoub (Reviewer #1); Jens Januschke (Reviewer #2).

The reviewers have discussed the reviews with one another and the Reviewing Editor has drafted this decision to help you prepare a revised submission. All reviewers agreed that the paper has been improved and many comments were satisfactorily addressed. However, there are some remaining points that need to be addressed, as outlined below:

Essential revisions:

1) The rescue experiment deserves some additional attention. The experimental setup used here shows that the RNAi line targets Alms1a, but it does not seem to conclusively show that only targets this gene. Alms1a-GFP in the RNAi background only partially restores the normal centrosome number of 2 centrosomes per cell. According to the data included in Figure 1—figure supplement 3, all control cells have 2 centrosomes, and in Alms1a RNAi, the large majority has zero and around 25% contain 1 centrosome. While Alms1a-GFP does not cause a change in centrosome number, in the RNAi background, only 25% of the cells contain 2 centrosomes. It is true that there is a decrease in the percentage of cells showing zero centrosomes, but this could be due to the excess of Alms1a-GFP and therefore reduced effect of the RNAi. Therefore, there was a concern that these data are insufficient to prove specificity of the RNAi used, and that the RNAi line is depleting other genes responsible for the phenotypes. Ideally, one would like to see characterization of another RNAi line or a mutation that can be fully rescued by the GFP line. Alternatively, one could generate a GFP line that it is not targeted by the RNAi. If this is not possible, the reviewers would like to see proof that in the rescue experiments, the depletion of the endogenous Alms1a not affected by the expression of Alms1a-GFP and some detailed discussion about the quality of the rescue experiment, which should get a more prominent place in the paper.

2) The point that Alms1a RNAi leads to the loss of the diplosome configuration of the GSC centrosomes is not demonstrated with sufficient clarity (if one looks closely there is a faint CNB signal at the mother centrosome). Therefore, it appears problematic to obtain sufficiently conclusive data with CNB as a marker. The authors should provide conclusive, quantified experimental evidence about centriole numbers per centrosome in GSCs upon knock down of Alsm1a (as in Figure 3F,G and Figure 2—figure supplement 2). The reviewers suggest using STED looking for cell cycle dependent ASL signal that allows visualizing the diplosome. The authors have now done this in controls (from where it appears that anaphase is suitable), but not in the relevant experimental conditions. These data should be included.

---

## [Author Response]

Essential revisions:1) The main observation is based on one RNAi line causing the phenotype. It would be important to add additional controls, a rescue or perhaps the use of a different RNAi line, or a mutant. In addition to the rescue with full-length protein, the authors can attempt the rescue experiment using the C-terminal fragment of the gene, since they suggest this is the key motif sufficient for interactions with both Klp10 at GSC centrosome, and for "recruiting" Sak during centriole duplication. Both points could be addressed with this one approach.

The rescue experiments have been performed and the results are shown in Figure 3—figure supplement 1. Full-length Alms1a protein is able to rescue the RNAi phenotype partially (possibly because of variable expression levels, whereby the expression level does not reach to a sufficient level in some cells/tissues). ALMS motif alone failed to rescue the RNAi phenotype. Even if this motif may be an essential domain to interact with Klp10A and Sak, it does not mean that this motif would be sufficient for Alms1a function(s) as the remaining region might mediate other essential functions such as linking Klp10A/Sak to something else.

2) Furthermore, while the central observation is remarkable, the study is somewhat preliminary in terms of the molecular mechanism. Refining the localization of Alms1a (and its C-terminal fragment) using a quantitative approach (potentially via STED or another superresolution imaging modality) would also strengthen the conclusion about the relative localization at mother vs daughter centrioles. At the same time, they can use this method to check the status of mother vs daughter centrioles in Alms1a-depleted cells.

Detailed characterization of Alms1a localization together with Cnb and Asl have been performed using STED (Figure 2P-R). Alms1a localizes to both mother and daughter centriole specifically in GSC mother centrosome as we stated. At the daughter centriole of the mother centrosome, it completely colocalizes with Cnb. We also included spermatocytes’ centrosome (Figure 2O) to show that Alms1a indeed localizes to the proximal side of the centriole.

The status of mother vs daughter centrioles in wild type and alms1RNAi GSCs has been examined with centrosome markers Cp110 and Asl using Lightning microscopy (Figure 3—figure supplement 2). Cp110 marks both mother and daughter centrioles in interphase wild type GSCs, consistent with previous studies that in GSCs centriole duplication occurs immediately after centrioles disengage. (Note that, this is in contrast to *Drosophila* neuroblast, where mother and daughter centrioles each organize centrosome without duplicating centrioles for quite some time during the cell cycle.) In contrast to wild type centrosomes, where two Cp110 dots were always observed, there is only one Cp110 spot on the elongated mother centrosome upon Alms1a depletion, suggesting that elongated centrosome is made of a single, elongated mother centriole, and there is no daughter centriole. Accordingly, many core centriole proteins and PCM proteins including Sak, Sas6, Ana1, Asl and g-Tub observed on “elongated mother centrosome” reflects their localization to the mother centriole (Figure 3 and 6).

3) Figure 6, the overexpression of Alms1 and Sak give rise to extra centrosomes even if Sak alone only results in a milder effect. This is very interesting as Sak over-expression is normally sufficient to generate extra centrosomes. However, the centrosomes in the panel Sak, Alms1 (Figure 6D) have different sizes and Asl intensities. Are the authors sure that they all represent extra bona fide centrosomes? Maybe using other centriole markers, such as Sas4, would be useful.

Additional markers including Sas-6 (centrioles) and Spd2 (both centrioles and PCM) have been used to characterize the centrosomes upon overexpression of Alms1a and Sak (Figure 6—figure supplement 1). Indeed, supernumerary centrosomes observed upon overexpression of Alms1a and Sak contains all centrosomal/centriolar markers, suggesting that these are bona fide centrosomes.

4) Finally, additional quantifications and controls (e.g. for the bipartide system to detect protein interactions) indicated in specific comments of the reviewers below would be important.

Quantifications and controls for BiFC have to added as shown in Figure 1—figure supplement 3 and Figure 5L. For threshold, we subtract the BiFC signal on ROI (centrosomes) to background signal (same size area in background), then use imageJ to quantify the integrated intensity density of IF signals.

We include below specific comments of individual reviewers for more background.Reviewer 1:1) The authors say "RNAi-mediated depletion of klp10A leads to abnormal elongation of the mother centrosomes in GSCs". Do they mean mother centriole? Or are both mother and daughter centrioles of the mother centrosome elongated? This also applies to Figure 3C-D, where the authors suggest "elongated mother centrosomes" upon Alms1 depletion. Based on the centriole duplication defect, I imagine they are referring to the existing mother centriole as being elongated.

In nos>klp10A^RNAi^ GSCs, only mother centriole elongates. Daughter centriole exists but not elongates (see Author response image 1, which is reproduced from Figure 2A-B in Chen et al., 2016 (published under the Creative Commons Attribution 4.0 International Public License, (CC BY 4.0, https://creativecommons.org/licenses/by/4.0/))). In nos>alms1^RNAi^ GSCs, only mother centriole exists and elongates (Figure 3—figure supplement 2). Also, Response 2 to Essential revision point 2 hopefully address this point further.

2) Figure 1E-F: The authors demonstrate interaction between Alms1a C-terminal motif with Klp10a, indicating this domain is sufficient. Did they test whether the N-terminus plays a role in this too? There are a number of centrosomal proteins that can interact via multiple domains.

We haven’t tested the function of Alms1a N-terminal motif yet. The reasons we expressed C-terminal domain only in some experiments reported here were 1) to test the function of ALMS motif (as it is the only conserved domain), and also 2) full length Alms1a does not express well in bacteria. Although it is possible that N-ter domain may also interact with other proteins, we feel that it is beyond the scope of the present study that identifies a protein that exhibits unique asymmetric localization on the GSC centrosomes for the first time.

3) Figure 2: Is the C-terminal domain of Alms1a, if expressed in the GSC, still show asymmetric association between mother and daughter centrosomes?

We did observe some asymmetric localization between mother and daughter centrosome in nos>GFP-ALMS motif GSCs (panel in Author response image 2). But we are not sure about the significance/meaning of this, and we feel that addressing such details is beyond the scope of the present study.

**Author response image 2. respfig2:** 

4) Figure 2—figure supplement1A: The authors show that klp10A-RNAi did not affect Alms1a localization to the SG centrosomes. Did they quantify the relative signal intensity between mother and daughter (since depletion of Klp10A does not cause mislocalization of Alms1a from centrosomes in GSCs completely either)?

Quantification have been added to the Figure 2—figure supplement 1.

5) Figure 5: Similar to Figure 1, they show interaction between Alms1a C-terminal motif with Sak, but not with Sas6. Did they identify interaction with Sas6 via the N-terminal half of Alms1a?

We haven’t done the GSC-pulldown assay to test the interaction of Sas6 with Alms1a N-ter. However, based on Bacterial Two Hybrid experiment (Figure 5—figure supplement 1), where we did not detect direct interaction between Sas6 and full-length Alms1a, there is no reason to believe Alms1a N-ter would interact with Sas-6.

6) Figure 6: The authors say "These results suggest that Alms1a promotes centrosome duplication together with Sak, possibly by recruiting Sak to the site of centriole duplication in GSCs". Do the authors know whether Alm1a is phosphorylated, and if so, is this Sak-dependent (since Sak's role in centriole duplication is typically mediated via phosphorylation)?

Based on the general knowledge of the centrosome field, it would not be surprising if Alms1a is phosphorylated by Sak. But we feel that this level of analysis is beyond the scope of the present study. We hope that the reviewers agree with our notion.

7) Figure 6: Importantly, if Alms1a's role in duplication is merely to recruit Sak, then one experiment to test that would be to express the C-terminal domain of Alms1a (which they show to be sufficient for interaction), to see if it can rescue the centriole duplication defect in Alms1a-depleted cells.

Please see our response to Essential revision 1.

Reviewer 2:1) In S-phase both the mother and daughter centrosomes of GSCs contain normally both a set of two centrioles. When centriole duplication is triggered again, both centrioles within the mother centrosome should be competent to duplicate. The data supporting that Alms1a regulates duplication of a specific type of centrioles are currently not strong. Cnb depletion in mother centrioles in GSCs is not quantified. Given that cnb mutant germ line cells do have centrioles despite basal body length regulation problems in spermatocytes (Reina, JCB 2018), it is not clear what depleted Cnb in the Alms1 RNAi GSCs means. While a possibility, it does not demonstrate per se that centriole duplication has not occurred. There is further no solid evidence to support that the mother centriole within the mother centrosome keeps duplicating when Alms1a is depleted, while the daughter centriole does not (appears also unlikely given the morphological changes (elongation) detected. The results point out that Alsm1a localizes to the proximal end of the mother centriole and does not overlap with Cnb, yet the discussion summarizes "Importantly, Alms1a's function is required in GSCs but not SGs, thus this GSC-specific localization at daughter centriole of the mother centrosome is likely what is required for centriole duplication." I got confused here and think that the rationale here needs to be better worked out.

We hope that the revision experiments (described above) have already clarified most of this reviewer’s concerns about the mode of centriole duplication in wild type and Alms1a-RNAi GSCs.

Quantification of Cnb staining has been added (new Figure 3H).

We would like to note that, unlike *Drosophila* neuroblasts, where single centriole (split mother and daughter centrioles) each organizes two centrosomes for quite some time during cell cycle before they eventually duplicate centrioles, male GSCs appear to duplicate centrioles as soon as mother and daughter centrioles split from each other during G1/S transition (Salzmann, 2014 MBoC). Noting this difference likely clarifies some concerns raised by this reviewer.

We believe that the lack of Cnb indicates the lack of daughter centrioles: please note the aforementioned centrosome composition in GSCs during cell cycle. Although this reviewer might have inferred that the lack of Cnb localization means that Cnb disappeared from a centriole, our data rather suggests that the daughter centriole to which Cnb can localize is not present. We hope the loss of daughter centriole is better supported now based on revision experiments (described above). We are not claiming that Alms1a regulates centriole duplication via Cnb—instead, we believe that the lack of Cnb simply demonstrates that there is no daughter centrioles (to which Cnb can localize).

2) Within the GSC centrosomes, would two rings of Asl in GSCs be expected normally in S-phase? This is what Figure 2C, D suggest. However, one Asl ring seems to be detected in Figure 2 F, G (SG cells) and in Figure 2O-Q (GSCs). It is possible that Asl labels the wall of only one centriole, despite two centrioles being present as at least in embryos Asl is only incorporated in the new daughter later in the centriole duplication cycle (e.g. Raff lab). Thorough quantifications of the localization of Asl signal (one or two centriole walls detected) is missing. This might be of relevance to assign centriole specific roles, especially as statements on the precise point of the cell and the centriole duplication cycle are tough to make in the system and could potentially be a confounding factor. Is there any way to control that rule that out? It is a pity that the STED analysis had not been extended to measure centrioles upon Alms1a depletion. STED on Asl, Alsm1a and Cnb in controls and upon Alms1 depletion in GSCs plus quantification might be desirable to better understand how loss of a specific pool/enrichment of Alms1 might drive a particular function at a given location.

In GSC centrosomes, Asl labels only the wall of mother centriole at interphase. Also the above-mentioned centrosome composition/centriole duplication timing in GSCs may now clarify some of confusions, and we hope that most of issues are clarified for this reviewer. In addition, STED analysis of the Asl localization has been performed and the results are shown as Figure 2—figure supplement 2.

3) Faulty centriole duplication is a likely possibility of centriole loss here, but centrioles could also be destabilized (fragmented and then lost). Can that be excluded? Especially since the gal80ts experiment revealed that after 4 days almost half of the GSCs appear to be centrosome less (Figure 3H model points at loss of centrosomes from GSCs), while in the nosGa4 driven depletion the conclusion is the GSCs specifically retain centrosomes (quantification missing). Or is mother centrosome anchoring depending on Alms1a? In my understanding the mother centrosome is "immortal" when properly anchored in this system.

The mother centrosome show ~95% of anchoring near the niche even in wild type, which means that the mother centrosome can be lost to differentiating cells at some frequency (not 100% “immortal”). After 4 days, which is 6-8 cell cycles, it is not surprising that significant population of GSCs have lost the centrosomes. Although the possibility of centriole destabilization may not be entirely excluded, given that the mother centrioles are stable enough (such that they can elongate so long) rather seems to favor the possibility of centriole loss. Also, we hope that the added revision experiments and explanations above clarify this reviewer’s concern on this point.

Reviewer 3:1) Alms1 RNAi. The authors show indeed a clear reduction by western blot. Did they rescue the depletion? Even if Alms RNAi results in lower levels, they should show that the phenotypes obtained are only due to this depletion. Using a resistant line or a mutation that can be rescued by Alms1 transgene will show this.

This issue was addressed in the Essential revision point 1.

2) Figure 2L, not sure I agree with the interpretation of Alms1 being localized to the proximal site of the two centrioles in the mother centrosome. Maybe super resolution will help and drawing line scans to illustrate the fluorescence intensity distribution?

The added revision experiments hopefully address this concern. Also, we added Alms1a localization in the spermatocytes, where much longer centrioles help visualize proximal vs. distal sides of centrioles (new Figure 2O). This clearly showed that Alms1a localizes at the proximal end of the centrioles.

3) Figure 5E, I am not sure I am interpreting this western blot correctly. While I can see in the IP for Sak the pull down of GST- Alms1 (Figure 5D), in the Sas-6 lane, this is not the case.

Correct. Sas-6 doesn’t interact with Alms1a C-ter, which is our conclusion (Sas-6 and Alms1a interacts in the GSC extract, but not via GST-pulldown using bacterial extract. Based on this, we concluded that Sas-6-Alms1a interaction is indirect).

4) The bipartide system to detect interactions- VNm9. I wonder if all the controls are included in the quantifications. Because in Figure 5 L the interaction in mother GSCs is more prominent, but still is not zero in the other centrosomes. Does it mean that they do not interact or that they interact with more molecules in the mother centrosome?

Quantifications of controls have been added in Figure 5L. There is no detectable BiFC signals in controls (both nos>alms1a-VNm9 and nos>Vc155-sak).

[Editors' note: further revisions were suggested prior to acceptance, as described below.]

Essential revisions:1) The rescue experiment deserves some additional attention. The experimental setup used here shows that the RNAi line targets Alms1a, but it does not seem to conclusively show that only targets this gene. Alms1a-GFP in the RNAi background only partially restores the normal centrosome number of 2 centrosomes per cell. According to the data included in Figure 1—figure supplement 3, all control cells have 2 centrosomes, and in Alms1a RNAi, the large majority has zero and around 25% contain 1 centrosome. While Alms1a-GFP does not cause a change in centrosome number, in the RNAi background, only 25% of the cells contain 2 centrosomes. It is true that there is a decrease in the percentage of cells showing zero centrosomes, but this could be due to the excess of Alms1a-GFP and therefore reduced effect of the RNAi. Therefore, there was a concern that these data are insufficient to prove specificity of the RNAi used, and that the RNAi line is depleting other genes responsible for the phenotypes. Ideally, one would like to see characterization of another RNAi line or a mutation that can be fully rescued by the GFP line. Alternatively, one could generate a GFP line that it is not targeted by the RNAi. If this is not possible, the reviewers would like to see proof that in the rescue experiments, the depletion of the endogenous Alms1a not affected by the expression of Alms1a-GFP and some detailed discussion about the quality of the rescue experiment, which should get a more prominent place in the paper.

We apologize for the confusion. We should have been clearer, but as was explained in the Materials and methods section, our rescue transgene is already RNAi-resistant. Therefore, it is highly unlikely that the introduction of transgene is weakening the RNAi effect, as the reviewers were concerned. Also, please note that the RNAi line that we used is from the “TRiP line” series that were designed and generated by Nobert Perrimon’s group, where the target sequence was designed computationally to avoid similarity to any other location of the *Drosophila* genome. The target sequence for this construct is TCGGAAGTAACTCAGACGAAA and upon BLAST search against *Drosophila* genome, we do not find any matching sequences except for Alms1a and Alms1b genes. Therefore, it is highly unlikely that the RNAi construct we used have unknown off-target effect (expect for targeting Alms1b additionally, because these two genes are almost identical).

Also, as explained in the manuscript, our attempts to generate any additional RNAi lines (only target Alms1a without targeting Alms1b) have been unsuccessful (we generated 2 lines but none of them efficiently targeted Alms1a).

Based on these information (in particular, the unlikeliness of off-target effects), our best explanation for partial rescue is the expression level of the transgene. It is widely known that nos-gal4 (the driver that we use for transgene expression) varies in expression level among cells. As necessary, we can add additional discussions in the main text to add caveats of our experiments.

Additional Editor comment for Essential revision 1Rescue: please make sure to explain in the main text how the rescue was done and include all the data pertaining to the rescue in the main figures. Please discuss all the various potential reasons explaining why the rescue is not complete (insufficient expression levels of the rescue construct, problems with the GFP fusion, etc). Since the rescue is incomplete, potential off-target effects of the RNAi remain a distinct possibility and must be included in the discussion - please soften your conclusions accordingly, and make sure that this possibility is very explicitly mentioned and will be obvious to the readers of the manuscript. With other words, you must clearly state in the manuscript that you cannot be sure that the phenotypes described are unambiguously linked to the lack of Alms1a function.

We have added the explanation in the main text in the Result section about these possibilities. With this addition of the text clearly describing caveats etc., we wish to keep the rescue data in the supplementary figures, simply because the data are too bulky to be included in the main figures. With a clear explanation in the text, this information (only partial rescue) is not hidden, and we believe that it is sufficient for transparency. On the contrary, we are concerned that the insertion of the bulky data into the main figure disrupts the flow of logic: there are many published papers whose entire organization had become very confusing and fragmented as a result of many revision processes, preventing the understanding by general readers. We believe that dissemination of scientific results must be mindful of broad scientific community (current and future), not just for authors and reviewers. By “flow” we do not mean the one that is convenient for us/authors, but the one better for the understanding by readers. If the editor/reviewers feel that the text dealing with experimental caveat is insufficient, we will further edit the text, but we hope that they agree that our treatment of this rescue data by the text provides sufficient transparency.

We also think that the question pertaining whether ALMS motif only might rescue the phenotype (which is included in this rescue figure) is rather too specific, and inclusion of such data into main figure does not benefit general readers. We added this experiment per reviewers’ request: if the result were to be positive, i.e. ALMS motif was sufficient to rescue the phenotype, it would have added an important knowledge, so we agreed with the reviewers’ point on the potential value of such an experiment. But now that we obtained the completely negative data, addition of such data in the main figure is not warranted. Again, inclusion of such data rather confuses readers, instead of providing better understanding and transparency.

2) The point that Alms1a RNAi leads to the loss of the diplosome configuration of the GSC centrosomes is not demonstrated with sufficient clarity (if one looks closely there is a faint CNB signal at the mother centrosome). Therefore, it appears problematic to obtain sufficiently conclusive data with CNB as a marker. The authors should provide conclusive, quantified experimental evidence about centriole numbers per centrosome in GSCs upon knock down of Alsm1a (as in Figure 3F,G and Figure 2—figure supplement 2). The reviewers suggest using STED looking for cell cycle dependent ASL signal that allows visualizing the diplosome. The authors have now done this in controls (from where it appears that anaphase is suitable), but not in the relevant experimental conditions. These data should be included.

We understand the potential concerns of reviewers on this point. However, we would like to bring their attention to the terminal phenotype of Alms1a-RNAi, as shown in Figure 3C, D, which is clear loss of daughter centrosomes. At this point, as is shown in Figure 3—figure supplement 2, the already-elongated mother centrosome contains only one Cp110 signal, as opposed to 2 dots/centrosome in the control (if deemed necessary, we can provide the quantification of Cp110 number per centrosome in control vs. Alms1a-RNAi GSCs). This clearly demonstrates that the final phenotype of Alms1a-RNAi is the loss of diplosome configuration. We believe that this is already sufficient to draw our conclusion that Alms1a is required for centriole duplication.

In this regard, Figure 3 F, G, H are only additional confirmation to our conclusion by capturing the moment of centrosome loss: when the mutant GSCs still contain 2 centrosomes = 2 g-tubulin dots (i.e. before GSCs exhibit the terminal phenotype of having elongated mother centrosome only), the mother centrosome already has a diminished amount (almost none) of Cnb---that’s the only point that we’re trying to convey, and these panels are not the only evidence to support our conclusion (because in the end, clearly the elongated mother centrosome does not have the diplosome configuration). We agree that one can see a tiny amount of Cnb still left on the mother centrosome, but this is clearly much less than the control situation (which has equal amount of Cnb on two centrosomes), and the fact that a tiny amount of Cnb is left on the mother centrosome does not negate our conclusion that the Alms1a-RNAi leads to “compromised daughter centriole generation”.

Moreover, as the data shown in Figure 3F-H represent “transitional phenotype” (instead of terminal phenotype), when the GSC is “about to lose the daughter centrosome”, capturing the right moment is extremely challenging. Anaphase cells are quite infrequent: Anaphase is <10% of entire mitotic cells, and GSCs’ overall mitotic index is only 2% (of total GSCs), this means that anaphase GSC is only less than 0.2% of entire GSCs. This is still feasible when we’re imaging wild type (Figure 2—figure supplement 2), but when combined with the need to capture the moment of centriole loss (specific time point when RNAi becomes effective), it becomes essentially impossible. Of course, lack of feasibility is not the reason for not needing the result, if the entire conclusion relies on that specific result, but as we outlined above, we believe that our overall conclusion (centriole duplication defect) is already sufficiently supported by Alms1a RNAi’s terminal phenotype (Figure 3—figure supplement 2).

We can certainly rephrase our main text to convey these points. We sincerely ask reviewers to consider these issues and hope they agree that we do not need additional experiments.

Additional Editor comment for Essential revision 2Conclusive, quantified experimental evidence about centriole numbers per centrosome in GSCs: indeed, please provide the quantification of Cp110 number per centrosome in control vs. Alms1a-RNAi GSCs. Overall, the reviewers have already raised the question about insufficient quantification in the first round of review and were somewhat disappointed by this aspect of the revised paper. Since CNB marker raises concerns, perhaps quantification would be appropriate as well? Again, please make sure that none of this information is "hidden" in the supplements.

We apologize if the addition of quantification data was not clear enough to reviewers in the first round of the revision. We had added the quantification of Cnb intensity (Figure 3H) per reviewers’ request during the first round of the revision, which shows a very low intensity of Cnb on the mother centrosome in Alms1-RNAi GSCs. In this revised version, we added more emphasis in the text on this aspect. Also Cp110 data was moved to the main Figure as new Figure 3I, J, with additional quantification as new Figure 3K.

In addition, as explained in our response letter that accompanied the first round of the revision, data quantification had been added to multiple places, addressing all of reviewers’ request during the first round of revision (Figure 5L, Figure 1—figure supplement 3, Figure 2—figure supplement 1B). Therefore, we are somewhat confused by the reviewers’ remarks that they were disappointed by the lack of quantification.